# Nearly Optimal Algorithms for Linear Contextual Bandits with Adversarial Corruptions

**Jiafan He**
Department of Computer Science
University of California, Los Angeles
jiafanhe19@ucla.edu

**Dongruo Zhou**
Department of Computer Science
University of California, Los Angeles
drzhou@cs.ucla.edu

**Tong Zhang**
Google Research & HKUST
tongzhang@tongzhang-ml.org

**Quanquan Gu**
Department of Computer Science
University of California, Los Angeles
qgu@cs.ucla.edu

## Abstract

We study the linear contextual bandit problem in the presence of adversarial corruption, where the reward at each round is corrupted by an adversary, and the corruption level (i.e., the sum of corruption magnitudes over the horizon) is $C \geq 0$. The best-known algorithms in this setting are limited in that they either are computationally inefficient or require a strong assumption on the corruption, or their regret is at least $C$ times worse than the regret without corruption. In this paper, to overcome these limitations, we propose a new algorithm based on the principle of optimism in the face of uncertainty. At the core of our algorithm is a weighted ridge regression where the weight of each chosen action depends on its confidence up to some threshold. We show that for both known $C$ and unknown $C$ cases, our algorithm with proper choice of hyperparameter achieves a regret that nearly matches the lower bounds. Thus, our algorithm is nearly optimal up to logarithmic factors for both cases. Notably, our algorithm achieves the near-optimal regret for both corrupted and uncorrupted cases ($C = 0$) simultaneously.

## 1 Introduction

We study linear contextual bandits with adversarial corruptions. At each round, the agent observes a decision set provided by the environment, and selects an action from the decision set. Then an adversary *corrupts* the reward of the action selected by the agent. The agent then receives the corrupted reward of the selected action and proceeds until $K$ rounds. The agent's goal is to minimize the regret Regret($K$), which is the difference between the optimal accumulated reward and the selected accumulated reward. This problem can be regarded as a combination of the two classical bandit problems, *stochastic bandits* and *adversarial bandits* (Lattimore and Szepesvári, 2018). In practice, the contextual bandits with adversarial corruptions can describe many popular decision-making problems such as pay-per-click advertisements with click fraud (Lykouris et al., 2018) and recommendation system with malicious users (Deshpande and Montanari, 2012).

Lykouris et al. (2018) first studied the multi-armed bandit with adversarial corruptions. Specifically, let $C$ denote the *corruption level* which is the sum of the corruption magnitudes at each round. Lykouris et al. (2018) proposed an algorithm with a regret that is $C$ times worse than the regret without corruption. Later, Gupta et al. (2019a) proposed an improved algorithm whose regret consists of two terms: a *corruption-independent* term that matches the optimal regret for multi-armed bandit without corruption, and a *corruption-dependent* term that is linear in $C$ and independent of $K$, i.e.,

Regret$(K) = o(K) + O(C)$. The lower bound proved in Gupta et al. (2019a) suggests that the linear dependence on $C$ is near-optimal. Such a regret structure reveals an desirable property of corruption-robust bandit algorithms, that is, the algorithm should perform nearly the same as the bandit algorithms without corruption when the corruption level $C$ is small or diminishes.

Based on the above observation, a natural question arises:

Can we design computationally efficient algorithms for linear contextual bandits with corruption that can attain the best possible regret, similar to those in multi-armed bandits?

Some previous works have attempted to answer this question for the simpler stochastic linear bandit setting, where the decision sets at each round are identical and finite. Li et al. (2019) studied the stochastic linear bandits and proposed an instance-dependent regret bound. Later on, Bogunovic et al. (2021) studied the same problem and proposed an algorithm that achieves a regret with the corruption term depending on $C$ linearly and on $K$ logarithmically. However, these algorithms are limited to the stochastic linear bandit setting since their algorithm design highly relies on the experiment design and arm-elimination techniques that require a multiple selection of the same action and can only handle fixed decision set. They are not applicable to contextual bandits, where the decision set is changing over time and can even be infinite. For the more general linear contextual bandit setting, Bogunovic et al. (2021) proved that a simple greedy algorithm based on linear regression can attain an ideal corruption term that has a linear dependence on $C$ and a logarithmic dependence on $K$, under a stringent diversity assumption on the contexts. Lee et al. (2021) proposed an algorithm and the corruption term in its regret depends on $C$ linearly and on $K$ logarithmically, but only holds for the restricted case when the corruption at each round is a linear function of the action. Without special assumptions on the contexts or corruptions, Zhao et al. (2021); Ding et al. (2021) proposed a variant of the OFUL algorithm (Abbasi-Yadkori et al., 2011) and its regret has a corruption term depending on $K$ polynomially. Recently, Wei et al. (2022) proposed a Robust VOFUL algorithm that achieves a regret with a corruption term linearly dependent on $C'$[1] and only logarithmically dependent on $K$. However, Robust VOFUL is computationally inefficient since it needs to solve a maximization problem over a nonconvex confidence set that is defined as the intersection of exponential number of sets, and its regret has a loose dependence on context dimension $d$. In addition, Wei et al. (2022) also proposed a Robust OFUL algorithm and provided a regret guarantee that has a linear dependence on a different notion of corruption level $C_r$[2], which is strictly larger than the corruption level $C$ considered in the previous work and the current paper. Thus, the above question remains open.

In this paper, we give an affirmative answer to the above question. We summarize our contributions as follows.

- We propose a computationally efficient algorithm based on the principle of optimism in the face of uncertainty (Abbasi-Yadkori et al., 2011), named Confidence-Weighted OFUL (CW-OFUL). At the core of our algorithm is a weighted ridge regression where the weight of each chosen arm is adaptive to its confidence, which is defined as the truncation of the inverse exploration bonus. Intuitively, such a weighting strategy prevents the algorithm from exploiting the contexts whose rewards are more likely corrupted by a large amount.

- For the case when the corruption level $C$ is known to the agent, we show that the proposed algorithm enjoys a regret Regret$(K) = \widetilde{O}(d\sqrt{K} + dC)$, where $d$ is the dimension of the contexts, $C$ is the corruption level and $K$ is the number of total iterations. The first term matches the regret lower bound of linear contextual bandits without corruption $\Omega(d\sqrt{K})$ (Lattimore and Szepesvári, 2018). The second term matches the lower bound on the corruption term in regret $\Omega(dC)$ (Bogunovic et al., 2021). They together suggest that our algorithm is not only robust but also near-optimal up to logarithmic factors.

- For the case when the corruption level $C$ is unknown to the agent, we show that CW-OFUL enjoys an $\widetilde{O}(d\sqrt{K})$ regret for the case $C \leq \sqrt{K}$, with proper choice of the hyperparameter. Surprisingly,

---

[1] In Wei et al. (2022), the adversary adds corruption to all actions in the decision set before observing the agent's action and they define the corruption level $C'$ as the maximum corruption over the decision set. See Remark 2.2 for the formal definition and a more detailed discussion.

[2] The corruption level $C_r$ is defined as $C_r = \sqrt{K \sum_{k=1}^{K} c_k^2}$, where $c_k \geq 0$ is the corruption magnitude at round $k$. As a comparison, $C = \sum_{k=1}^{K} |c_k|$. In the worst case, $C_r = O(\sqrt{K}C)$ and therefore the corruption term in the regret of Robust OFUL will depend on $K$ polynomially.

by proving a lower bound on the regret, we show that our regret upper bound is already optimal for all algorithms that achieve a near-optimal regret bound for uncorrupted bandits.

We compare our regret bounds with previous ones in Table 1. We can see that our algorithm matches the lower bound up to logarithmic factors in both known $C$ and unknown $C$ cases, and therefore is nearly optimal.

Table 1: Comparisons of regrets for corrupted linear contextual bandits.

| Algorithm | Regret | $C$ | Efficiency [3] | Adversary [4] |
|---|---|---|---|---|
| Robust weighted OFUL (Zhao et al., 2021) | $\widetilde{O}(d\sqrt{K} + dC'\sqrt{K})$ | Known | Yes | Weak |
| Robust OFUL (Wei et al., 2022) | $\widetilde{O}(d\sqrt{K} + C_r)$ | Known | Yes | Weak |
| Robust VOFUL (Wei et al., 2022) | $\widetilde{O}(d^{4.5}\sqrt{K} + d^4 C')$ | Known | No | Weak |
| CW-OFUL (Theorem 4.2) | $\widetilde{O}(d\sqrt{K} + dC)$ | Known | Yes | Strong |
| CW-OFUL (Remark 4.4) | $\widetilde{O}(d\sqrt{K} + dC')$ | Known | Yes | Weak |
| Lower bound (Lattimore and Szepesvári, 2018) (Bogunovic et al., 2021) | $\Omega(d\sqrt{K} + dC)$ | Known | N/A | Strong |
| Multi-level weighted OFUL (Zhao et al., 2021) | $\widetilde{O}(dC'^2\sqrt{K}),\ C' = \Omega(1)$ | Unknown | Yes | Weak |
| Greedy (Bogunovic et al., 2021) | $\widetilde{O}\big((\sqrt{dK} + C)/\lambda_0\big)$ [5] | Unknown | Yes | Strong |
| COBE+OFUL (Wei et al., 2022) | $\widetilde{O}(d\sqrt{K} + C_r)$ | Unknown | Yes | Weak |
| COBE+VOFUL (Wei et al., 2022) | $\widetilde{O}(d^{4.5}\sqrt{K} + d^4 C')$ | Unknown | No | Weak |
| CW-OFUL($\bar{C} = \sqrt{K}$) (Theorem 4.9) | $\widetilde{O}(d\sqrt{K}),\ C \leq \sqrt{K}$ $O(K),\ C \geq \sqrt{K}$ | Unknown | Yes | Strong |
| COBE + CW-OFUL (Remark 4.10 | $\widetilde{O}(d\sqrt{K} + dC')$ | Unknown | Yes | Weak |
| Lower bound[6] ( Lattimore and Szepesvári 2018) (Theorem 4.12) | $\Omega(d\sqrt{K}),\ C \leq \sqrt{K}$ $\Omega(K),\ C \geq \sqrt{K}$ | Unknown | N/A | Strong |

## 1.1 Additional Related Work

**Bandits with Misspecification.** Bandits with misspecification can be seen as a special case of bandit with adversarial corruption since it is corrupted relatively evenly at each round. Let $\epsilon$ be the misspecification level. Ghosh et al. (2017) firstly studied the stochastic linear bandits and proved a sublinear regret when $\epsilon$ is small. Lattimore and Szepesvari (2019) studied the stochastic linear bandit

---

[3]The weak adversary must corrupt the rewards before the agent selects its actions, while the powerful adversary (i.e., strong adversary) can corrupt the rewards after seeing the action being selected by the agent.

[4]In this work, we assume there is a computation oracle to solve the linear optimization problems over the decision set $\mathcal{D}_t$ (e.g., Line 3 of Algorithm 1). This is implicitly assumed in almost all existing works for solving contextual linear bandit problems with infinite arms (e.g., OFUL and LinUCB algorithms); otherwise, choosing an arm from the infinite decision set is computationally intractable. In the special case that the decision set is finite or the convex hull of a finite set, such a computation oracle apparently exists. ).

[5]Greedy Bogunovic et al. (2021) assumes that each arm in the decision set at each round is sampled from a distribution that satisfies $(r, \lambda_0)$-diverse property (Kannan et al., 2018) . A distribution $\mathcal{D}$ is $(r, \lambda_0)$-diverse if for any $\mathbf{a} = \boldsymbol{\mu} + \boldsymbol{\xi}$ with $\boldsymbol{\mu} \in \mathbb{R}^d$ and $\boldsymbol{\xi} \sim \mathcal{D}$, $\lambda_{\min}(\mathbb{E}_{\boldsymbol{\xi} \in \mathcal{D}}[\mathbf{a}\mathbf{a}^\top | \boldsymbol{\theta}^\top \boldsymbol{\xi} \geq b]) \geq \lambda_0$ holds for all $\boldsymbol{\theta} \in \mathbb{R}^d$ and $b \in \mathbb{R}$ satisfying $b \leq r\|\boldsymbol{\theta}\|_2$.

[6]The lower bound under a large corruption level $C \geq \sqrt{K}$ only holds for algorithms that can achieve near-optimal regret for uncorrupted bandits. It is possible for an algorithm that does not achieve the optimal regret for uncorrupted bandits (e.g., $R_K = O(K^{0.75})$) to achieve a sub-linear regret in the presence of corruptions.

setting under milder assumptions. With the knowledge of $\epsilon$, they proposed an algorithm with an $\widetilde{O}(\sqrt{dK\log(N)}+\epsilon\sqrt{dK})$ regret, where $d$ is the dimension of the contextual vector, $N$ is the number of arms. Their regret bound matches their proved lower bound up to logarithmic factors. Foster et al. (2020) further considered the more general linear contextual bandits with misspecification when $\epsilon$ is unknown to the agent, and proposed an algorithm equipped with a CORRAL meta algorithm (Agarwal et al., 2017) to deal with the unknown $\epsilon$. Their algorithm enjoys an $\widetilde{O}(d\sqrt{K}+\epsilon\sqrt{dK})$ regret. Krishnamurthy et al. (2021) proposed an algorithm without using a meta algorithm which has the same order of regret as Foster et al. (2020). Our algorithm can be directly applied to the misspecification setting by choosing the corruption level $C$ to be $K\epsilon$, which immediately gives us an $\widetilde{O}(d\sqrt{K}+dK\epsilon)$ regret upper bound.

**Bandits with Adversarial Rewards.** There exists a large body of literature on the problems of adversarial multi-armed bandits (Auer et al., 2002; Bubeck and Cesa-Bianchi, 2012). There is also a line of works trying to design algorithms that can achieve near-optimal regret bounds for both stochastic bandits and adversarial bandits simultaneously (Bubeck and Slivkins, 2012; Seldin and Slivkins, 2014; Auer and Chiang, 2016; Seldin and Lugosi, 2017; Zimmert and Seldin, 2019; Lee et al., 2021). However, most of these algorithms focus on the general adversarial reward setting without specifying the total amount of corruption. One of the notable exceptions is Lee et al. (2021), which assumed that the adversarial corruptions are generated through the inner product of an adversarial vectors and the contextual vector. As a comparison, our algorithm and result do not need such additional assumption on the structure of the corruption. Our algorithm can be applied to both corrupted and uncorrupted settings with different choices of hyperparameters, and achieves a near-optimal regret for both cases.

**Notation** We use lower case letters to denote scalars, and use lower and upper case bold face letters to denote vectors and matrices respectively. We denote by $[n]$ the set $\{1,\ldots,n\}$. For a vector $\mathbf{x}\in\mathbb{R}^d$ and a positive semi-definite matrix $\mathbf{\Sigma}\in\mathbb{R}^{d\times d}$, we denote by $\|\mathbf{x}\|_2$ the vector's $\ell_2$ norm and by $\|\mathbf{x}\|_{\mathbf{\Sigma}}=\sqrt{\mathbf{x}^\top\mathbf{\Sigma}\mathbf{x}}$ the Mahalanobis norm. For two positive sequences $\{a_n\}$ and $\{b_n\}$ with $n=1,2,\ldots$, we write $a_n=O(b_n)$ if there exists an absolute constant $C>0$ such that $a_n\le Cb_n$ holds for all $n\ge 1$ and write $a_n=\Omega(b_n)$ if there exists an absolute constant $C>0$ such that $a_n\ge Cb_n$ holds for all $n\ge 1$. We use $\widetilde{O}(\cdot)$ to further hide the polylogarithmic factors. We use $\mathbb{1}\{\cdot\}$ to denote the indicator function.

## 2 Preliminaries

In this section, we introduce the setting of linear contextual bandit with adversarial corruption.

**Linear contextual bandit with corruption.** We define linear contextual bandits with corruption as follows: at the beginning of each round $k\in[K]$, the agent receives a decision set $\mathcal{D}_k\subseteq\mathbb{R}^d$ from the environment and it chooses an action (i.e., arm, contextual vector) $\mathbf{x}\in\mathcal{D}_k$. After choosing the action $\mathbf{x}_k$ at round $k$, the environment generates the corresponding $r_k'$ based on the stochastic linear model $r_k'=\langle\boldsymbol{\theta}^*,\mathbf{x}\rangle+\eta_k$, where $\boldsymbol{\theta}^*\in\mathbb{R}^d$ is an unknown environment parameter and $\eta_k$ is the stochastic noise. After seeing the stochastic reward $r_k'$, the adversary (i.e., attacker) introduces an adversarial corruption $c_k$ onto the reward, which may depend on the decision set $\mathcal{D}_k$, action $\mathbf{x}_k$, stochastic reward $r_k'$. Finally, the agent observes the corrupted reward $r_k=\langle\boldsymbol{\theta}^*,\mathbf{x}\rangle+\eta_k+c_k$ at round $k$. Following Abbasi-Yadkori et al. (2011), we make the following assumptions on the bandit model.

**Assumption 2.1.** The linear contextual bandit satisfies the following conditions:

- At each round $k$ and any action $\mathbf{x}\in\mathcal{D}_k$, we have $\|\mathbf{x}\|_2\le L$.

- For the unknown environment parameter $\boldsymbol{\theta}^*$, it satisfies $\|\boldsymbol{\theta}^*\|_2\le S$.

- At each round $k$, the corresponding stochastic noise $\eta_k$ is conditional $R$-sub-Gaussian, i.e.,

$$\forall\lambda\in\mathbb{R},\ \mathbb{E}\big[e^{\lambda\eta_k}|\mathbf{x}_{1:k},\eta_{1:k-1},c_{1:k-1}\big]\le\exp(R^2\lambda^2/2).$$

**Regret.** The goal of the agent is to minimize the pseudo-regret in the first $K$ rounds, which is defined as follows:

$$\text{Regret}(K)=\sum_{k=1}^K\max_{\mathbf{x}\in\mathcal{D}_k}\langle\boldsymbol{\theta}^*,\mathbf{x}\rangle-\langle\boldsymbol{\theta}^*,\mathbf{x}_k\rangle.$$

**Corruption level.** To measure the level of adversarial corruptions, we define the *corruption level* as $C := \sum_{k=1}^{K} |c_k|$. With this definition, we say a linear contextual bandit problem is $C$-corrupted if and only if the corruption level is no larger than $C$.

**Remark 2.2.** The adversary in our setting and the corresponding definition of corruption level is the same as that in Bogunovic et al. (2021) and slightly different from that in prior works such as Lykouris et al. (2018); Gupta et al. (2019b); Zhao et al. (2021). More specifically, in these works, the adversarial corruption $c_k$ is chosen before the choice of action $\mathbf{x}_k \in \mathcal{D}_k$. Since the actions selected by the agent may not be deterministic, the adversary chooses different corruption $c_{k,\mathbf{x}}$ for different action $\mathbf{x} \in \mathcal{D}_k$. With this notion of corruption, the corresponding corruption level is defined as $C' = \sum_{k=1}^{K} \max_{\mathbf{x} \in \mathcal{D}_k} |c_{k,\mathbf{x}}|$. As a comparison, our adversary chooses the corruption after observing the action $x_k$ and for the corruption level. We have

$$C = \sum_{k=1}^{K} |c_{k,\mathbf{x}_k}| \le \sum_{k=1}^{K} \max_{\mathbf{x} \in \mathcal{D}_k} |c_{k,\mathbf{x}}| = C',$$

which implies that our corruption level $C$ is always no larger than the corruption level $C'$ in Lykouris et al. (2018); Gupta et al. (2019b); Zhao et al. (2021).

## 3 Algorithms

In this section, we review existing algorithms for linear contextual bandits (and stochastic linear bandits) and discuss their limitations when they are applied to the adversarial corruption setting. Then we present our algorithm CW-OFUL and illustrate how our algorithm design can overcome the these limitations.

### 3.1 Existing Algorithms

We begin with reviewing the classical OFUL algorithm (Abbasi-Yadkori et al., 2011). Under Assumption 2.1, at round $k$, OFUL estimates $\boldsymbol{\theta}^*$ by online ridge regression over all the past observed actions and rewards, i.e.,

$$\boldsymbol{\theta}_k \leftarrow \underset{\boldsymbol{\theta} \in \mathbb{R}^d}{\operatorname{argmin}} \lambda \|\boldsymbol{\theta}\|_2^2 + \sum_{i=1}^{k-1} \left( \boldsymbol{\theta}^\top \mathbf{x}_i - r_i \right)^2. \tag{3.1}$$

With $\boldsymbol{\theta}_k$ in hand, OFUL constructs a confidence set for $\boldsymbol{\theta}^*$ as follows $\mathcal{C}_k = \left\{ \boldsymbol{\theta} : \|\boldsymbol{\theta}_k - \boldsymbol{\theta}\|_{\boldsymbol{\Sigma}_k} \le \beta \right\}$, where $\beta$ is the confidence radius and $\boldsymbol{\Sigma}_k = \lambda \mathbf{I} + \sum_{i=1}^{k-1} \mathbf{x}_i \mathbf{x}_i^\top$ is the covariance matrix of contexts $\mathbf{x}_i, i = 1, \ldots, k$. Without corruption, it is known that setting $\beta = \widetilde{O}(R\sqrt{d})$ guarantees that $\boldsymbol{\theta}^* \in \mathcal{C}_k$ with high probability, which further leads to a sublinear regret $\widetilde{O}(d\sqrt{K})$. However, with corruption, such a choice of $\beta$ is not sufficient. To see why, we take a closer look at the closed-form solution $\boldsymbol{\theta}_k$ to (3.1):

$$\boldsymbol{\theta}_k = \boldsymbol{\Sigma}_k^{-1} \sum_{i=1}^{k-1} \mathbf{x}_i r_i = \boldsymbol{\Sigma}_k^{-1} \sum_{i=1}^{k-1} \mathbf{x}_i (\mathbf{x}_i^\top \boldsymbol{\theta}^* + \eta_i) + \boldsymbol{\Sigma}_k^{-1} \sum_{i=1}^{k-1} \mathbf{x}_i c_i.$$

By simple calculation and assuming $\lambda$ to be a constant, we can show that $\|\boldsymbol{\theta}_k - \boldsymbol{\theta}^*\|_{\boldsymbol{\Sigma}_k}$ can be upper bounded by

$$\|\boldsymbol{\theta}_k - \boldsymbol{\theta}^*\|_{\boldsymbol{\Sigma}_k} \le O\left( \underbrace{\left\| \sum_{i=1}^{k-1} \mathbf{x}_i \eta_i \right\|_{\boldsymbol{\Sigma}_k^{-1}}}_{I_1} + \underbrace{\left\| \sum_{i=1}^{k-1} \mathbf{x}_i c_i \right\|_{\boldsymbol{\Sigma}_k^{-1}}}_{I_2} \right).$$

The first term $I_1$ is corruption-independent and bounded by $\widetilde{O}(R\sqrt{d})$ according to Abbasi-Yadkori et al. (2011). The challenge is to bound the second term $I_2$, which depends on the corruption. Existing approaches (Zhao et al., 2021; Ding et al., 2021) bound $I_2$ by triangle inequality and Cauchy-Schwarz inequality,

$$I_2 \le \sum_{i=1}^{k-1} \|\mathbf{x}_i c_i\|_{\boldsymbol{\Sigma}_k^{-1}} \le \sum_{i=1}^{k-1} |c_i| \max_{1 \le j \le k-1} \|\mathbf{x}_j\|_{\boldsymbol{\Sigma}_k^{-1}} \le \sum_{i=1}^{k-1} |c_i| L/\sqrt{\lambda} = O(C), \tag{3.2}$$

---

**Algorithm 1** CW-OFUL

---

**Require:** Regularization parameter $\lambda$, confidence radius $\beta$ and threshold parameter $\alpha$
1: **for** round $k = 1, 2, ..$ **do**
2:     Set $\mathbf{\Sigma}_k = \lambda\mathbf{I} + \sum_{i=1}^{k-1} w_i \mathbf{x}_i \mathbf{x}_i^\top$
3:     Set $\mathbf{b}_k = \sum_{i=1}^{k-1} w_i \mathbf{x}_i r_i$ and $\boldsymbol{\theta}_k = \mathbf{\Sigma}_k^{-1}\mathbf{b}_k$
4:     Receive the decision set $\mathcal{D}_k$
5:     Choose action $\mathbf{x}_k \leftarrow \text{argmax}_{\mathbf{x}\in\mathcal{D}_k} \boldsymbol{\theta}_k^\top \mathbf{x} + \beta\sqrt{\mathbf{x}^\top \mathbf{\Sigma}_k^{-1}\mathbf{x}}$
6:     Set $w_k = \min\{1, \alpha/\|\mathbf{x}_k\|_{\mathbf{\Sigma}_k^{-1}}\}$
7: **end for**

---

where $C$ is the corruption level and $\max_{1\leq j\leq k-1} \|\mathbf{x}_j\|_{\mathbf{\Sigma}_k^{-1}}$ is bounded by the crude upper bound $L/\sqrt{\lambda}$. Unfortunately, such a bound makes the confidence radius be the order of $O(R\sqrt{d} + C)$, which eventually leads to an term $O(C\sqrt{K})$ in the regret, which is $C$ times worse than the regret without corruption.

In order to obtain a tighter bound of $I_2$, for stochastic linear bandits, Bogunovic et al. (2021) proposed a Robust Phase Elimination (RPE) algorithm, which employs *optimal design* (Lattimore and Szepesvári, 2018) to select the arms. In this setting, the decision set is finite and fixed over time, i.e., $\mathcal{D}_k = \mathcal{D}$ for all $k \in [K]$ and $|\mathcal{D}| \leq \infty$. More specifically, RPE divides the time horizon into several phases. Within each phase, RPE performs linear regression on a multiset $\mathcal{A} \subset \mathcal{D}$, which is the G-optimal design of $\mathcal{D}$. Here the multiset means $\mathcal{A}$ has duplicate elements. Let $\mathbf{\Sigma}$ be the covariance matrix defined over $\mathcal{A}$, then the following upper bound holds (Lattimore and Szepesvári, 2018):

$$\forall \mathbf{x} \in \mathcal{D}, \ \|\mathbf{x}\|_{\mathbf{\Sigma}^{-1}} = O\big(|\mathcal{A}|^{-1/2}\big). \tag{3.3}$$

By choosing a large enough $|\mathcal{A}|$, (3.3) provides a *uniformly small* upper bound for $\max_{1\leq j\leq k-1} \|\mathbf{x}_j\|_{\mathbf{\Sigma}_k^{-1}}$ for any $k$. Substituting (3.3) back into (3.2) with $|\mathcal{A}| = O(C^2)$, we can show that $I_2$ is bounded by some small constant, which therefore eliminates the $O(C\sqrt{K})$ term in the final regret. Although the optimal design-based approach RPE (Bogunovic et al., 2021) successfully eliminates the multiplicative term $C\sqrt{K}$, it is not applicable to our linear contextual bandit setting: (1) it needs to select a *multiset* from the decision set, which is impossible for the general contextual bandit setting; (2) the complexity of optimal design introduces some additional quadratic term $C^2$ in their final regret, which makes their algorithm non-optimal (See Bogunovic et al. (2021) for more details).

### 3.2 Our Algorithm

As we have seen before, it is pivotal to bound the corruption-dependent term $I_2$ tightly. To overcome the limitations of existing approaches, we propose a fundamentally new approach and present our CW-OFUL in Algorithm 1. At a high level, Algorithm 1 is an extension of the OFUL algorithm (Abbasi-Yadkori et al., 2011), which is also based on the principle of optimism in the face of uncertainty.

Our algorithm assigns a weight $w_k$ to each selected action $\mathbf{x}_k$. More specifically, at round $k$, we use the following weighted ridge regression to estimate the unknown vector $\boldsymbol{\theta}^*$:

$$\boldsymbol{\theta}_k \leftarrow \underset{\boldsymbol{\theta}\in\mathbb{R}^d}{\text{argmin}} \ \lambda\|\boldsymbol{\theta}\|_2^2 + \sum_{i=1}^{k-1} w_i\big(\boldsymbol{\theta}^\top \mathbf{x}_i - r_i\big)^2. \tag{3.4}$$

The closed-form solution to the above optimization problem is displayed in Line 3 of Algorithm 1. While weighted ridge regression is not new and has been used in prior work on bandits (Kirschner and Krause, 2018; Zhou et al., 2021; Russac et al., 2019), the setting, motivation and the choice of weight are fundamentally different. More specifically, we choose the weight as the *truncation* of the inverse exploration bonus, which is $w_k = \min\left\{1, \alpha/\|\mathbf{x}_k\|_{\mathbf{\Sigma}_k^{-1}}\right\}$. Here $\alpha > 0$ is a threshold parameter. We can see that for action $\mathbf{x}_k$ with a large exploration bonus $\|\mathbf{x}_k\|_{\mathbf{\Sigma}_k^{-1}}$ (low confidence), CW-OFUL will assign a small weight to it to avoid the potentially large regret caused by both the stochastic noise and the adversarial corruption. On the other hand, for the action with a small exploration bonus (high

confidence), CW-OFUL will assign a large weight to it (it can be as large as 1). Another interesting observation is that by setting $\alpha$ to be sufficiently large, the weight will become 1 for every action, and CW-OFUL will degenerate to OFUL (Abbasi-Yadkori et al., 2011).

As a comparison, Kirschner and Krause (2018); Zhou et al. (2021) used the inverse of the noise variance as the weight to normalize the noise and derived tight variance-dependent regret guarantees. Russac et al. (2019) set the weight as a geometric sequence to perform moving average to deal with the non-stationary environment.

To see how our choice of weight can lead to tighter regret, we first write down the closed-form solution to (3.4)

$$\boldsymbol{\theta}_k = \boldsymbol{\Sigma}_k^{-1} \sum_{i=1}^{k-1} w_i \mathbf{x}_i (\mathbf{x}_i^\top \boldsymbol{\theta}^* + \eta_i) + \sum_{i=1}^{k-1} \boldsymbol{\Sigma}_k^{-1} w_i \mathbf{x}_i c_i,$$

where the covariance matrix $\boldsymbol{\Sigma}_k = \lambda \mathbf{I} + \sum_{i=1}^{k-1} w_i \mathbf{x}_i \mathbf{x}_i^\top$. With some calculation and assuming $\lambda$ to be a constant, we can obtain

$$\|\boldsymbol{\theta}_k - \boldsymbol{\theta}^*\|_{\boldsymbol{\Sigma}_k} \le O\bigg( \underbrace{\bigg\| \sum_{i=1}^{k-1} w_i \mathbf{x}_i \eta_i \bigg\|_{\boldsymbol{\Sigma}_k^{-1}}}_{I_1} + \underbrace{\bigg\| \sum_{i=1}^{k-1} w_i \mathbf{x}_i c_i \bigg\|_{\boldsymbol{\Sigma}_k^{-1}}}_{I_2} \bigg).$$

$I_1$ is the corruption-independent term and can still be bounded by $\widetilde{O}(R\sqrt{d})$ according to Abbasi-Yadkori et al. (2011). For $I_2$, we have

$$\bigg\| \sum_{i=1}^{k-1} w_i \mathbf{x}_i c_i \bigg\|_{\boldsymbol{\Sigma}_k^{-1}} \le \sum_{i=1}^{k-1} |c_i| w_i \|\mathbf{x}_i\|_{\boldsymbol{\Sigma}_k^{-1}} \le \sum_{i=1}^{k-1} |c_i| \alpha = C\alpha,$$

It is evident that with our carefully designed weight, the corruption-dependent term $I_2$ can be uniformly bounded by some constant $C\alpha$, the same as that in Bogunovic et al. (2021). Therefore, by setting $\alpha$ to be sufficiently small, our CW-OFUL can get rid of the $C\sqrt{K}$ term in the final regret.

## 4 Main Results

In this section, we present the main theoretical guarantees of CW-OFUL.

### 4.1 Known Corruption Level $C$: Upper Bound

We first consider the case when $C$ is known to the agent. In this case, we choose $\alpha = R\sqrt{d}/C$. The following lemma characterizes the estimation error of $\boldsymbol{\theta}_k$ with respect to $\boldsymbol{\theta}^*$, which is a formal summary of our discussion in Section 3.

**Lemma 4.1.** Suppose that Assumption 2.1 holds. For any $0 < \delta < 1$ and corruption budget $C \ge 0$, set the confidence radius $\beta = R\sqrt{d \log\left((1 + KL^2/\lambda)/\delta\right)} + \sqrt{\lambda}S + \alpha C$ in Algorithm 1, then with probability at least $1 - \delta$, for every round $k$, the estimator $\boldsymbol{\theta}_k$ satisfies that $\|\boldsymbol{\theta}_k - \boldsymbol{\theta}^*\|_{\boldsymbol{\Sigma}_k} \le \beta$.

The following theorem provides the regret bound of Algorithm 1.

**Theorem 4.2.** Suppose that Assumption 2.1 holds. For any $0 < \delta < 1$ and corruption budget $C \ge 0$, set the confidence radius $\beta$ in Algorithm 1 as follows:

$$\beta = R\sqrt{d \log\left((1 + KL^2/\lambda)/\delta\right)} + \alpha C + \sqrt{\lambda}S.$$

Then with probability at least $1 - \delta$, its regret in the first $K$ rounds is upper bounded by

$$\text{Regret}(K) = O\bigg( dR\sqrt{K \log^2\left((1 + KL^2/\lambda)/\delta\right)} + \alpha C\sqrt{dK \log^2\left((1 + KL^2/\lambda)/\delta\right)}$$

$$+ S\sqrt{d\lambda K \log(1 + KL^2/\lambda)} + \frac{Rd^{1.5}}{\alpha} \times \sqrt{\log^3\left((1 + KL^2/\lambda)/\delta\right)} \bigg)$$

$$+ \frac{dS\sqrt{\lambda}}{\alpha} \times \sqrt{\log^2\left((1+KL^2/\lambda)/\delta\right)} + dC\sqrt{\log^2\left((1+KL^2/\lambda)/\delta\right)}\Bigg).$$

In addition, if choosing $\alpha = (R\sqrt{d} + \sqrt{\lambda}S)/C$ and $\lambda = R^2/S^2$, its regret can be upper bounded by

$$\text{Regret}(K) = \widetilde{O}(d\sqrt{K} + dC).$$

A few remarks about Theorem 4.2 are in order.

**Remark 4.3.** Compared with the $\widetilde{O}(d\sqrt{K} + dC\sqrt{K})$ regret proved in Zhao et al. (2021); Ding et al. (2021), our algorithm improves the multiplicative dependence on corruption level $C$ to additive dependence. In particular, CW-OFUL achieves the same order of regret as the uncorrupted setting when $C = O(\sqrt{K})$, and it attains a sublinear regret as long as $C = o(K)$. In sharp contrast, the algorithm proposed in Zhao et al. (2021) achieves the same order of regret as the uncorrupted setting only when $C = O(1)$, and has a sublinear regret only when $C = o(\sqrt{K})$.

**Remark 4.4.** We also compare our result with that in Wei et al. (2022). The Robust+OFUL algorithm in Wei et al. (2022) achieves an $\widetilde{O}(d\sqrt{K} + C_r)$ regret with $C_r = \sqrt{T\sum_{k=1}^{K} c_k^2}$, which will degenerate to $\widetilde{O}(d\sqrt{K} + d\sqrt{K}C)$ in the worst case. Their regret guarantee is always worse than ours when $C < \sqrt{K}$. In addition, according to the discussion in Remark 2.2, Theorem 4.2 also implies an $\widetilde{O}(d\sqrt{K} + dC')$ regret under the notion of the corruption level $C'$. In contrast, the Robust VOFUL algorithm in Wei et al. (2022) has an $\widetilde{O}(d^{4.5}\sqrt{K} + d^4 C')$ regret, which is also inferior to our regret. Furthermore, Robust VOFUL is computationally inefficient.

**Remark 4.5.** We further compare our result with previous additive regrets derived for stochastic linear bandits. Let $\mathcal{D}_k = \mathcal{D}$ be the decision set. Compared with the $O(\sqrt{dK\log|\mathcal{D}|} + Cd^{3/2})$ regret for stochastic linear bandit with corruption derived in Bogunovic et al. (2021), our regret improves the corruption term by a factor of $\sqrt{d}$. Note that the $\sqrt{d}$ difference in the leading $\sqrt{K}$ term between our regret and theirs is caused by the fact that Bogunovic et al. (2021) considered the finite-arm setting, while we consider the infinite-arm setting. Our algorithm will have the same regret as theirs when $|\mathcal{D}| = O(\exp(d))$.

**Remark 4.6.** For the uncorrupted setting where $C = 0$, Theorem 4.2 suggests that the threshold parameter $\alpha$ should be set to infinity. Then by Line 6 in Algorithm 1, each weight $w_k$ becomes 1, and CW-OFUL degenerates to OFUL. Meanwhile, the regret in Theorem 1 also becomes $\widetilde{O}(d\sqrt{K})$ that matches the regret of OFUL (Abbasi-Yadkori et al., 2011).

## 4.2 Known Corruption Level C: Lower Bound

In this subsection, we refer to two existing lower bound results to show that when $C$ is known, our $\widetilde{O}(d\sqrt{K} + dC)$ regret is optimal up to logarithmic factors. The first proposition shows that the $\widetilde{O}(d\sqrt{K})$ corruption-independent term in our regret is near-optimal.

**Proposition 4.7** (Theorem 24.2, Lattimore and Szepesvári 2018). Assume $d \leq 2K$, $R = 1$ and $\mathcal{D}_k = \{\|\mathbf{x}\|_2 \leq 1\}$ for all $k \geq 1$. Then for any algorithm, there exists an environment parameter vector $\boldsymbol{\theta}^* \in \mathbb{R}^d$ satisfying $\|\boldsymbol{\theta}^*\|_2^2 = d^2/(48K)$ such that $\mathbb{E}(\text{Regret}(K)) \geq d\sqrt{K}/(16\sqrt{3})$.

The second proposition suggests that the $O(dC)$ corruption term in our regret is optimal.

**Proposition 4.8** (Theorem 3, Bogunovic et al. 2021). For any dimension $d$, for any algorithm that has the knowledge of $C$, there exists an instance satisfying with probability at least 0.5, $\text{Regret}(K) = \Omega(dC)$.

Combining Propositions 4.7 and 4.8, we can conclude that for any algorithm, there exists a corrupted bandit instance such that the algorithm suffers at least $\Omega(\max\{d\sqrt{K}, dC\})$ regret. Such a lower bound matches our upper bound up to logarithmic factors. Therefore, our algorithm is nearly optimal.

## 4.3 Unknown Corruption Level C: Upper Bound

Now we consider the case when $C$ is unknown. Our solution is quite simple for this case: we introduce a tuning parameter $\bar{C}$, which can be viewed as an estimate of $C$, and select the threshold

parameter $\alpha$ as Theorem 4.2 suggests. The following theorem gives the regret upper bound of CW-OFUL for the unknown $C$ case.

**Theorem 4.9.** Under the same conditions of Theorem 4.2 except that we set $\alpha = (R\sqrt{d} + \sqrt{\lambda}S)/\bar{C}$ with $\bar{C}$ being an estimated corruption level, $\lambda = R^2/S^2$ and $\beta = 2R\sqrt{d\log\big((1 + KL^2/\lambda)/\delta\big)} + 2\sqrt{\lambda}S$ in Algorithm 1. The regret of CW-OFUL can be upper bounded in the following two cases:

- If the corruption level $C$ satisfies that $0 \leq C \leq \bar{C}$, then with probability at least $1 - \delta$, the regret is upper bounded by $\text{Regret}(K) = \widetilde{O}(dR\sqrt{K} + d\bar{C})$.

- If the corruption level $C$ satisfies that $C > \bar{C}$, the regret is upper bounded by $\text{Regret}(K) = O(K)$.

In addition, if we set the estimation $\bar{C} = \sqrt{K}$, then when $0 \leq C \leq \sqrt{K}$, the regret is upper bounded by $\widetilde{O}(d\sqrt{K})$.

**Remark 4.10.** Zhao et al. (2021) proposed an $\widetilde{O}(C^2 d\sqrt{K})$ regret with unknown $C = \Omega(1)$. Compared with their result, our regret (with $\bar{C} = \sqrt{K}$) is strictly better in the corruption term. Bogunovic et al. (2021) proposed an $\widetilde{O}(\sqrt{dK\log|\mathcal{D}|} + Cd^{1.5} + C^2)$ regret for the stochastic linear bandit with unknown $C$, in the regime $C = \widetilde{O}(\sqrt{K}/d)$, where $\mathcal{D}$ is the finite decision set. Such a regret becomes $\widetilde{O}(d\sqrt{K} + Cd^{1.5} + C^2)$ when the size of $\mathcal{D}$ becomes exponentially large in $d$ or even infinite. Compared with their regret, our regret is not only smaller, but also holds for a wider regime (i.e., $C = O(\sqrt{K})$). Compared with the greedy algorithm in Bogunovic et al. (2021), our result does not rely on the stringent $(r, \lambda_0)$-diverse property assumption on the contexts.

**Remark 4.11.** We also compare our result (choosing $\bar{C} = \sqrt{K}$) with those in Wei et al. (2022) for the unknown $C$ case. Wei et al. (2022) proposed a COBE+OFUL algorithm with an $\widetilde{O}(d\sqrt{K} + C_r)$ regret, and a COBE+VOFUL algorithm with an $\widetilde{O}(d^{4.5}\sqrt{K} + d^4 C')$ regret, analogous to their results for the known $C$ case discussed in Remark 4.4. Our CW-OFUL enjoys a better regret than COBE+OFUL for all $C$, and it is better than COBE+VOFUL for $C < \sqrt{K}$. In addition, for the modified notion of corruption level $C'$, if we choose the basic algorithm in COBE (Wei et al., 2022) as our CW-OFUL algorithm, then Theorem 3 in Wei et al. (2022) suggests that COBE+CW-OFUL can deal with unknown corruption level $C'$ and obtained an $\widetilde{O}(d\sqrt{K} + dC')$ regret guarantee, which matches the regret of CW-OFUL algorithm with known corruption level $C'$. Note that COBE+VOFUL is also computationally inefficient.

### 4.4  Unknown Corruption Level $C$: Lower Bound

With $\bar{C} = \sqrt{K}$, for the case when $0 \leq C \leq \sqrt{K}$, our regret result is already near-optimal, due to the lower bound for the uncorrupted bandit in Proposition 4.7. Now we show that our $O(K)$ bound, seemingly trivial, is actually optimal for a large class of bandit algorithms. In detail, the following theorem provides a lower bound result for any algorithm for the unknown $C$ case. This is an extension of the lower bound result in Bogunovic et al. (2021) from $d = 2$ to general $d$.

**Theorem 4.12.** For any algorithm **Alg**, let $R_K$ be an upper bound of $\text{Regret}(K)$ such that for any bandit instance satisfying Assumption 2.1 with $C = 0$, it satisfies the $\mathbb{E}\big[\text{Regret}(K)\big] \leq R_K \leq O(K)$, where the expectation is with respect to the randomness of the algorithm and the stochastic noise. Then for the general case with $C = \Omega(R_K/d)$, such an algorithm will have $\mathbb{E}\big[\text{Regret}(K)\big] = \Omega(K)$.

**Remark 4.13.** If we selects the estimated corruption $\bar{C} = \Omega(R_K/d)$, Theorem 4.9 immediately implies that CW-OFUL enjoys a $O(R_K)$ regret when corruption level $C < \Omega(R_K/d)$ and $O(K)$ regret when corruption level $C \geq \Omega(R_K/d)$. Compared with the algorithm **Alg**, Theorem 4.12 suggests that CW-OFUL is no worse than the algorithm **Alg** no matter whether the corruption level $C < \Omega(R_K/d)$. More discussion can be found in Appendix A.1.

## 5  Conclusion and Future Work

In this work, we study corrupted linear contextual bandits. We propose a CW-OFUL algorithm based on a weighted ridge regression with truncated inverse exploration bonus weights. We show that for

both cases when the corruption level $C$ is known or unknown to the agent, CW-OFUL achieves a regret that matches the lower bound up to logarithmic factors.

We are also interested in achieving the optimal regret when specializing our algorithm to the misspecified linear contextual bandits.

## Acknowledgments and Disclosure of Funding

We thank the anonymous reviewers and area chair for their helpful comments. JH, DZ and QG are supported in part by the National Science Foundation CAREER Award 1906169 and the Sloan Research Fellowship. TZ is supported in part by the GRF 16201320. The views and conclusions contained in this paper are those of the authors and should not be interpreted as representing any funding agencies.

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
