# A  Additional Results

## A.1  Discussion on the Lower Bound for Unknown Corruption Level $C$

Consider a class of algorithms $\mathcal{A}$ whose worst-case regret is $R_K$ in the uncorrupted case. Here we only need to consider $\Omega(d\sqrt{K}) \leq R_K \leq O(K)$, since for any algorithm, $\Omega(d\sqrt{K})$ is the lowest possible worst-case regret (Lattimore and Szepesvári, 2018) and $O(K)$ is the highest possible regret. We first show that CW-OFUL belongs to $\mathcal{A}$. Choosing $\bar{C} = R_K/d$, Theorem 4.9 immediately suggests that CW-OFUL enjoys a $R_K$ regret in the uncorrupted case (i.e., $C = 0$). Thus CW-OFUL belongs to $\mathcal{A}$. Then we will show that CW-OFUL is the best possible one in $\mathcal{A}$. On the one hand, Theorem 4.9 suggests that CW-OFUL suffers a linear regret when $C > \bar{C} = R_K/d$. On the other hand, Theorem 4.12 shows that any algorithm with $R_K$ regret in the uncorrupted case should have a linear regret when $C = \Omega(R_K/d)$. These together imply that CW-OFUL is optimal within $\mathcal{A}$.

## A.2  Discussion on the Misspecified Linear Bandits

We consider the misspecified linear bandit setting which assumes that the corruption at each round is uniformly bounded by $\epsilon$. Clearly, the misspecified linear bandit is a special case of corrupted linear contextual bandit with $C = K\epsilon$. Theorem 4.2 suggests that a direct application of our algorithm to this special setting incurs an $\widetilde{O}(d\sqrt{K} + dK\epsilon)$ regret, which differs from the near-optimal regret $\widetilde{O}(d\sqrt{K} + \sqrt{d}K\epsilon)$ (Lattimore and Szepesvari, 2019; Foster et al., 2020) by a $\sqrt{d}$ factor on the corruption term. Whether our algorithm is able to achieve the near-optimal regret for both misspecified linear bandit and corrupted linear contextual bandit simultaneously remains an open question.

# B  Experiments

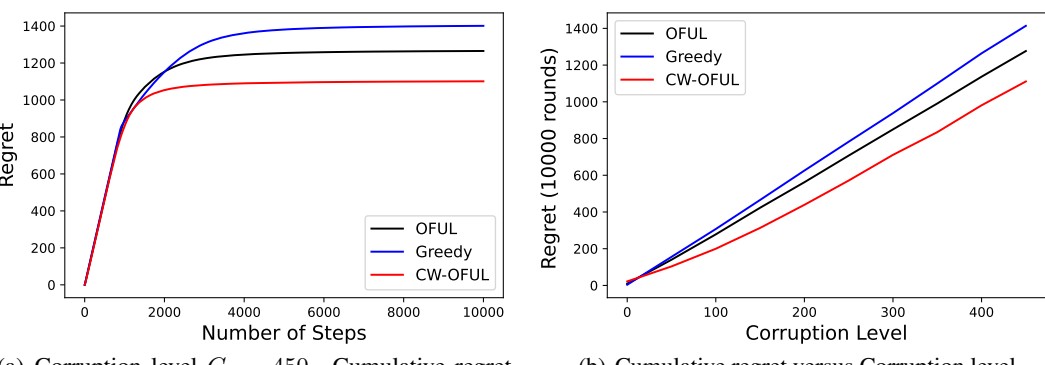

(a) Corruption level $C = 450$. Cumulative regret versus Round

(b) Cumulative regret versus Corruption level

Figure 1: Comparison of CW-OFUL(ours), Greedy (Bogunovic et al., 2021) and OFUL(Abbasi-Yadkori et al., 2011). Experiments are run for unknown corruption levels $C$ from $\{0, 50, 100, ..., 450\}$ (10 different corruption levels) , and results are averaged over 100 runs. Figure 1(a) presents the cumulative regrets with unknown corruption $C = 450$ ; Figure 1(b) shows the cumulative regret versus unknown corruption level $C$.

In this section, we run experiments and evaluate the performance of our algorithm CW-OFUL with an unknown corruption level $C$, which corroborates our theory. The code and data for our experiments can be found on Github [3].

**Model Parameters** We construct a linear bandit instance with dimension $d = 5$ and the true model parameter $\boldsymbol{\theta}^*$ is denoted by

$$\boldsymbol{\theta}^* = \left[\frac{1}{\sqrt{d}}, ..., \frac{1}{\sqrt{d}}\right]^\top \in \mathbb{R}^d.$$

---

[3]https://github.com/uclaml/CW-OFUL

During each round $k \in [K]$, the decision set $\mathcal{D}_k$ consists of 20 different actions and each action is uniformly sampled from the space $[+1/\sqrt{d}, -1/\sqrt{d}]^d$, which satisfies the positive minimum eigenvalue assumption for greedy algorithm (Bogunovic et al., 2021). In addition, after choosing the action, $\mathbf{x}_k$ in round $k \in [K]$, an 0.1-Gaussian noise $\eta_k$ will be added to the reward.

**Attack method** For the attack method, we choose the flip-$\boldsymbol{\theta}$ attack. More specifically, with a corruption level of $C$, the adversary tricks the learner by flipping the value, i.e., $r_t(\mathbf{x}_t) = -\langle \mathbf{x}_t, \theta^* \rangle + \eta_k$ in the first $C$ rounds. In the remaining rounds, the adversary does not corrupt the reward.

**Results and discussions** In our experiments, we make a simulation with the total number of rounds $K = 10000$ (repeating 100 times and taking the average) and corruption levels from $\{0, 50, 100, ..., 450\}$ (10 different corruption levels ). We applied our CW-OFUL algorithm and compared its performance with greedy (Bogunovic et al., 2021) and OFUL (Abbasi-Yadkori et al., 2011).

The experimental results are shown in Figure 1 These simulation results suggest that our CW-OFUL algorithm outperforms both the Greedy and OFUL algorithms. Specifically, Figure 1(a) displays the cumulative regret of our algorithm, OFUL, and Greedy algorithm under the same number of rounds. They show that our algorithm slightly outperforms them in terms of regret.

Figure 1(b) plots the cumulative regret versus the unknown corruption level. We can see that all of the three algorithms demonstrate an additive linear dependence on the unknown corruption level $C$, which corroborate our theoretical guarantee.

# C   Instance-dependent Regrets

Prior works (Lykouris et al., 2018; Li et al., 2019; Zhao et al., 2021) have proved instance-dependent regret bounds for corruption-robust linear bandits. We show that CW-OFUL also enjoys an instance-dependent regret bound. Following Abbasi-Yadkori et al. (2011), we define the minimal sub-optimality gap as follows.

**Definition C.1** (Minimal sub-optimality gap)**.** For each round $k \in [K]$ and any action $\mathbf{x} \in \mathcal{D}_k$, the sub-optimality gap $\Delta_{\mathbf{x},k}$ is defined as

$$\Delta_{\mathbf{x},k} = \max_{\mathbf{x}^* \in \mathcal{D}_k} \langle \boldsymbol{\theta}^*, \mathbf{x}^* \rangle - \langle \boldsymbol{\theta}^*, \mathbf{x} \rangle,$$

and the minimal sub-optimality gap is defined as

$$\Delta = \min_{k \in [K], \mathbf{x} \in \mathcal{D}_k} \left\{ \Delta_{\mathbf{x},k} : \Delta_{\mathbf{x},k} \neq 0 \right\}. \tag{C.1}$$

We assume that the minimal sub-optimality gap is strictly positive.

**Assumption C.2.** The minimal sub-optimality gap is strictly positive, i.e., $\Delta > 0$.

Under the assumption of positive minimal sub-optimality, the following theorem provides an instance-dependent regret guarantee for CW-OFUL.

**Theorem C.3.** Under the same conditions of Theorem 4.2, with high probability at least $1 - \delta$, the regret of Algorithm 1 in the first $K$ rounds is upper bounded by

$$\text{Regret}(K) \leq O\bigg( R^2 d^2 \log^2 \big((1 + KL^2/\lambda)/\delta\big)/\Delta + \frac{\alpha^2 dC^2}{\Delta} \times \sqrt{\log\big(3 + C^2 L^2 K/(R^2\lambda\delta)\big)}$$

$$+ S^2 d\lambda \log(1 + KL^2/\lambda)/\Delta + \frac{Rd^{1.5}}{\alpha} \times \sqrt{\log^3\big((1 + KL^2/\lambda)/\delta\big)}$$

$$+ \frac{dS\sqrt{\lambda}}{\alpha} \times \sqrt{\log^2\big((1 + KL^2/\lambda)/\delta\big)} + dC\sqrt{\log^2\big((1 + KL^2/\lambda)/\delta\big)} \bigg)$$

In addition, if choosing $\alpha = (R\sqrt{d} + \sqrt{\lambda}S)/C$ and $\lambda = R^2/S^2$, the regret can be upper bounded by

$$\text{Regret}(K) \leq \widetilde{O}(d^2/\Delta + dC).$$

**Remark C.4.** Our regret is strictly better than the $\widetilde{O}(d^{2.5}C/\Delta + d^6/\Delta^2)$ regret proved by Li et al. (2019) under a stronger assumption. Meanwhile, Zhao et al. (2021) implies an $\widetilde{O}(d^2C/\Delta)$ regret for their algorithm under the known $C$ case, which is also worse than our result.

## D  Overview of Key Proof Techniques

In this section, we give an overview of the main technical difficulty and our proof technique to derive Theorem 4.2.

By the standard regret decomposition technique from Abbasi-Yadkori et al. (2011), we upper bound the regret by the sum of the exploration bonuses times the confidence radius:

$$\text{Regret}(K) = O\left(\beta \cdot \sum_{k=1}^{K} \sqrt{\mathbf{x}_k^\top \boldsymbol{\Sigma}_k^{-1} \mathbf{x}_k}\right). \tag{D.1}$$

Lemma 4.1 suggests $\beta \sim R\sqrt{d} + \alpha C$. Therefore, we only need to bound the summation of the exploration bonuses. For the basic case when $w_k = 1$, we bound it using the elliptical potential lemma (Abbasi-Yadkori et al., 2011) as follows

$$\sum_{w_k=1} \sqrt{\mathbf{x}_k^\top \boldsymbol{\Sigma}_k^{-1} \mathbf{x}_k} \leq \sum_{k=1}^{K} \sqrt{\mathbf{x}_k^\top \left(\lambda \mathbf{I} + \sum_{i=1}^{k-1} \mathbf{x}_i \mathbf{x}_i^\top\right)^{-1} \mathbf{x}_k} \sim \widetilde{O}(\sqrt{dK}), \tag{D.2}$$

which contributes to the corruption-independent term $dR\sqrt{K}$ in our regret. For the case when $w_k < 1$, however, we are facing the *weighted* covariance matrix and cannot directly use the elliptical potential lemma. A trivial approach is to lower bound the weights by their *uniform* lower bound, i.e.,

$$\lambda \mathbf{I} + \sum_{i=1}^{k-1} w_i \mathbf{x}_i \mathbf{x}_i^\top \succeq \min_{1 \leq i \leq k-1} w_i \cdot \left(\lambda \mathbf{I} + \sum_{i=1}^{k-1} \mathbf{x}_i \mathbf{x}_i^\top\right). \tag{D.3}$$

By the definition of the weight $w_i$ in Algorithm 1 and a crude upper bound for the exploration bonus, we conclude from the definition of $w_k$ that $w_k = \Omega(\alpha)$. Substituting it into (D.3), we only obtain a regret $\widetilde{O}(\sqrt{dK}/\alpha)$, which is not satisfying.

To overcome this issue, we recall the definition for weight $w_k < 1$ in Algorithm 1: $w_k = \alpha/\|\mathbf{x}_k\|_{\boldsymbol{\Sigma}_k^{-1}}$ and we can bound the summation of the exploration bonuses as

$$\sum_{w_k<1} \sqrt{\mathbf{x}_k^\top \boldsymbol{\Sigma}_k^{-1} \mathbf{x}_k} = \sum_{k=1}^{K} w_k \mathbf{x}_k^\top \boldsymbol{\Sigma}_k^{-1} \mathbf{x}_k/\alpha \sim \widetilde{O}(d/\alpha). \tag{D.4}$$

Combining the results in (D.2) and (D.4) into (D.1), we can prove the final regret.

## E  Proof of Theorem 4.2

In this section, we provide the proof of Theorem 4.2. For simplicity, we use $\mathcal{E}$ to denote the following event:

$$\mathcal{E} = \left\{\|\boldsymbol{\theta}_k - \boldsymbol{\theta}^*\|_{\boldsymbol{\Sigma}_k} \leq \beta, \forall k \in [K]\right\}.$$

Lemma 4.1 shows that $\Pr(\mathcal{E}) \geq 1 - \delta$.

**Lemma E.1.** If setting the confidence radius $\beta = R\sqrt{d \log\left((1 + KL^2/\lambda)/\delta\right)} + \alpha C + \sqrt{\lambda}S$ in Algorithm 1, then on the event $\mathcal{E}$, for each round $k \in [K]$, the regret at round $k$ is upper bounded by

$$\Delta_k = \max_{\mathbf{x} \in \mathcal{D}_k} \langle \boldsymbol{\theta}^*, \mathbf{x} \rangle - \langle \boldsymbol{\theta}^*, \mathbf{x}_k \rangle \leq 2\beta \sqrt{\mathbf{x}_k^\top \boldsymbol{\Sigma}_k^{-1} \mathbf{x}_k}.$$

*Proof of Theorem 4.2.* Based on the event $\mathcal{E}$, the regret in the first $K$ round can be decomposed into two parts based on the weight $w_k$:

$$\text{Regret}(K) = \sum_{k=1}^{K} \max_{\mathbf{x} \in \mathcal{D}_k} \langle \boldsymbol{\theta}^*, \mathbf{x} \rangle - \langle \boldsymbol{\theta}^*, \mathbf{x}_k \rangle$$

$$\leq \min\left(2, \sum_{k=1}^{K} 2\beta\sqrt{\mathbf{x}_k^\top \boldsymbol{\Sigma}_k^{-1}\mathbf{x}_k}\right)$$

$$= \underbrace{\sum_{k:w_k=1} \min\left(2, 2\beta\sqrt{\mathbf{x}_k^\top \boldsymbol{\Sigma}_k^{-1}\mathbf{x}_k}\right)}_{I_1} + \underbrace{\sum_{k:w_k<1} \min\left(2, 2\beta\sqrt{\mathbf{x}_k^\top \boldsymbol{\Sigma}_k^{-1}\mathbf{x}_k}\right)}_{I_2}, \qquad \text{(E.1)}$$

where the inequality holds due to the Lemma E.1 with the fact that the suboptimality in each round $k$ is no more than 2.

For the term $I_1$, we consider for all rounds $k \in [K]$ with $w_k = 1$ and we assume these rounds can be listed as $\{k_1, .., k_m\}$ for simplicity. With this notation, for each $i \leq m$, we can construct the auxiliary covariance matrix $\mathbf{A}_i = \lambda\mathbf{I} + \sum_{j=1}^{i-1} \mathbf{x}_{k_j}\mathbf{x}_{k_j}^\top$. Due to the definition of original covariance matrix $\boldsymbol{\Sigma}_k$ in Algorithm (Line 2), we have

$$\boldsymbol{\Sigma}_{k_i} \geq \lambda\mathbf{I} + \sum_{j=1}^{i-1} w_{k_j}\mathbf{x}_{k_j}\mathbf{x}_{k_j}^\top = \mathbf{A}_i.$$

According to Lemma J.4, it further implies that for vector $\mathbf{x}_{k_i}$, we have

$$\mathbf{x}_{k_i}^\top \boldsymbol{\Sigma}_{k_i}^{-1}\mathbf{x}_{k_i} \leq \mathbf{x}_{k_i}^\top (\mathbf{A}_i)^{-1}\mathbf{x}_{k_i}. \qquad \text{(E.2)}$$

Therefore, the term $I_1$ can be bounded by

$$I_1 = \sum_{k:w_k=1} \min\left(2, 2\beta\sqrt{\mathbf{x}_k^\top \boldsymbol{\Sigma}_k^{-1}\mathbf{x}_k}\right)$$

$$\leq \sum_{i=1}^{m} 2\beta\min\left(1, \sqrt{\mathbf{x}_{k_i}^\top \boldsymbol{\Sigma}_{k_i}^{-1}\mathbf{x}_{k_i}}\right)$$

$$\leq 2\beta\sum_{i=1}^{m} \min\left(1, \sqrt{\mathbf{x}_{k_i}^\top (\mathbf{A}_i)^{-1}\mathbf{x}_{k_i}}\right)$$

$$\leq 2\beta\sqrt{\sum_{i=1}^{m} 1 \times \sum_{i=1}^{m} \min\left(1, \mathbf{x}_{k_i}^\top (\mathbf{A}_i)^{-1}\mathbf{x}_{k_i}\right)}$$

$$\leq 2\beta\sqrt{2dK\log(1 + KL^2/\lambda)}, \qquad \text{(E.3)}$$

where the first inequality holds since $\beta \geq 1$, the second inequality holds due to (E.2), the third inequality holds due to Cauchy-Schwarz inequality, the last inequality holds due to Lemma J.3 with the facts that $m \leq K$ and $\|\mathbf{x}_{k_i}\|_2 \leq L$.

For the second term $I_2$, according to the definition for weight $w_k < 1$ in Algorithm 1, we have $w_k = \alpha/\sqrt{\mathbf{x}_k^\top \boldsymbol{\Sigma}_k^{-1}\mathbf{x}_k}$, which implies that

$$I_2 = \sum_{k:w_k<1} \min\left(2, 2\beta\sqrt{\mathbf{x}_k^\top \boldsymbol{\Sigma}_k^{-1}\mathbf{x}_k}\right)$$

$$= \sum_{k:w_k<1} \min\left(2, 2\beta w_k\mathbf{x}_k^\top \boldsymbol{\Sigma}_k^{-1}\mathbf{x}_k/\alpha\right)$$

$$\leq \sum_{k:w_k<1} \min\left((2 + 2\beta/\alpha), (2 + 2\beta/\alpha)w_k\mathbf{x}_k^\top \boldsymbol{\Sigma}_k^{-1}\mathbf{x}_k\right)$$

$$= \sum_{k:w_k<1} (2 + 2\beta/\alpha)\min\left(1, w_k\mathbf{x}_k^\top \boldsymbol{\Sigma}_k^{-1}\mathbf{x}_k\right), \qquad \text{(E.4)}$$

where the second equation holds due to the definition of weight $w_k$. Now, we assume the rounds with weight $w_k < 1$ can be listed as $\{k_1, .., k_m\}$ for simplicity. In addition, we introduce the auxiliary vector $\mathbf{x}_i'$ as $\mathbf{x}_i' = \sqrt{w_{k_i}}\mathbf{x}_{k_i}$ and matrix $\boldsymbol{\Sigma}_i'$ as

$$\boldsymbol{\Sigma}_i' = \lambda\mathbf{I} + \sum_{j=1}^{i-1} w_{k_j}\mathbf{x}_{k_j}\mathbf{x}_{k_j}^\top = \lambda\mathbf{I} + \sum_{j=1}^{i-1} \mathbf{x}_j'(\mathbf{x}_j')^\top.$$

According to Lemma J.4, we have $(\mathbf{\Sigma}'_i)^{-1} \succeq \mathbf{\Sigma}_{k_i}^{-1}$. Therefore, for each $i \in [m]$, we have

$$\mathbf{x}_{k_i}^\top (\mathbf{\Sigma}'_i)^{-1} \mathbf{x}_{k_i} \geq \mathbf{x}_{k_i}^\top \mathbf{\Sigma}_{k_i}^{-1} \mathbf{x}_{k_i}, \tag{E.5}$$

where the inequality holds due to $(\mathbf{\Sigma}'_i)^{-1} \succeq \mathbf{\Sigma}_{k_i}^{-1}$. Now, taking a summation of (E.5) over all rounds $k_i$, we have

$$\sum_{i=1}^m \min\left(1, w_{k_i} \mathbf{x}_{k_i}^\top \mathbf{\Sigma}_{k_i}^{-1} \mathbf{x}_{k_i}\right) \leq \sum_{i=1}^m \min\left(1, w_{k_i} \mathbf{x}_{k_i}^\top (\mathbf{\Sigma}'_i)^{-1} \mathbf{x}_{k_i}\right)$$

$$= \sum_{i=1}^m \min\left(1, (\mathbf{x}'_i)^\top (\mathbf{\Sigma}'_i)^{-1} \mathbf{x}'_i\right)$$

$$\leq 2d \log(1 + KL^2/\lambda), \tag{E.6}$$

where the first inequality holds due to (E.5), the second inequality holds due to Lemma J.3 with the facts that $m \leq K$. Substituting the result in (E.6) into (E.4), the term $I_2$ can be upper bounded by

$$I_2 \leq \sum_{k:w_k<1} (2 + 2\beta/\alpha) \min\left(1, w_k \mathbf{x}_k^\top \mathbf{\Sigma}_k^{-1} \mathbf{x}_k\right)$$

$$\leq (2 + 2\beta/\alpha) \times 2d \log(1 + KL^2/\lambda). \tag{E.7}$$

Finally, substituting the results in (E.3) and (E.7) into (E.1), the regret can be upper bounded by

$$\text{Regret}(K) \leq 2\beta\sqrt{2dK \log(1 + KL^2/\lambda)} + (2 + 2\beta/\alpha) \times 2d \log(1 + KL^2/\lambda)$$

$$= O\left(dR\sqrt{K \log^2\left((1 + KL^2/\lambda)/\delta\right)} + \alpha C\sqrt{dK \log^2\left((1 + KL^2/\lambda)/\delta\right)}\right.$$

$$+ S\sqrt{d\lambda K \log(1 + KL^2/\lambda)} + \frac{Rd^{1.5}}{\alpha} \times \sqrt{\log^3\left((1 + KL^2/\lambda)/\delta\right)}$$

$$\left. + \frac{dS\sqrt{\lambda}}{\alpha} \times \sqrt{\log^2\left((1 + KL^2/\lambda)/\delta\right)} + dC\sqrt{\log^2\left((1 + KL^2/\lambda)/\delta\right)}\right).$$

Therefore, we complete the proof of Theorem 4.2. $\qquad\square$

## F  Proof of Theorem C.3

In this section, we present the detailed proof of Theorem 4.2.

*Proof of Theorem C.3.* Based on the event $\mathcal{E}$, the regret in round $k \in [K]$ is upper bounded by

$$\Delta_k = \max_{\mathbf{x} \in \mathcal{D}_k} \langle \boldsymbol{\theta}^*, \mathbf{x} \rangle - \langle \boldsymbol{\theta}^*, \mathbf{x}_k \rangle \leq 2\beta\sqrt{\mathbf{x}_k^\top \mathbf{\Sigma}_k^{-1} \mathbf{x}_k}.$$

On the other hand, according to Assumption C.2, the regret in round $k \in [K]$ satisfies that $\Delta_k = 0$ or $\Delta_k \geq \Delta$. Combining these two results, for round $k \in [K]$ with uncertainty $2\beta\sqrt{\mathbf{x}_k^\top \mathbf{\Sigma}_k^{-1} \mathbf{x}_k} < \Delta$, the regret must satisfy $\Delta_k = 0$. Therefore, the regret in the first $K$ rounds can be decomposed to two part based on the weight $w_k$ and exploration bonus $\sqrt{\mathbf{x}_k^\top \mathbf{\Sigma}_k^{-1} \mathbf{x}_k}$:

$$\text{Regret}(K) = \sum_{k=1}^K \max_{\mathbf{x} \in \mathcal{D}_k} \langle \boldsymbol{\theta}^*, \mathbf{x} \rangle - \langle \boldsymbol{\theta}^*, \mathbf{x}_k \rangle$$

$$= \sum_{k:2\beta\sqrt{\mathbf{x}_k^\top \mathbf{\Sigma}_k^{-1} \mathbf{x}_k} \geq \Delta} \max_{\mathbf{x} \in \mathcal{D}_k} \langle \boldsymbol{\theta}^*, \mathbf{x} \rangle - \langle \boldsymbol{\theta}^*, \mathbf{x}_k \rangle$$

$$\leq \min\left(2, \sum_{k:2\beta\sqrt{\mathbf{x}_k^\top \mathbf{\Sigma}_k^{-1} \mathbf{x}_k} \geq \Delta} 2\beta\sqrt{\mathbf{x}_k^\top \mathbf{\Sigma}_k^{-1} \mathbf{x}_k}\right)$$

$$= \sum_{k:w_k=1, 2\beta\sqrt{\mathbf{x}_k^\top \boldsymbol{\Sigma}_k^{-1}\mathbf{x}_k} \geq \Delta} \min\left(2, 2\beta\sqrt{\mathbf{x}_k^\top \boldsymbol{\Sigma}_k^{-1}\mathbf{x}_k}\right)$$

$$+ \sum_{k:w_k<1, 2\beta\sqrt{\mathbf{x}_k^\top \boldsymbol{\Sigma}_k^{-1}\mathbf{x}_k} \geq \Delta} \min\left(2, 2\beta\sqrt{\mathbf{x}_k^\top \boldsymbol{\Sigma}_k^{-1}\mathbf{x}_k}\right)$$

$$\leq \underbrace{\sum_{k:w_k=1, k:2\beta\sqrt{\mathbf{x}_k^\top \boldsymbol{\Sigma}_k^{-1}\mathbf{x}_k} \geq \Delta} \min\left(2, 2\beta\sqrt{\mathbf{x}_k^\top \boldsymbol{\Sigma}_k^{-1}\mathbf{x}_k}\right)}_{J_1}$$

$$+ \underbrace{\sum_{k:w_k<1} \min\left(2, 2\beta\sqrt{\mathbf{x}_k^\top \boldsymbol{\Sigma}_k^{-1}\mathbf{x}_k}\right)}_{J_2}, \tag{F.1}$$

where the inequality holds due to Lemma E.1 with the fact that the suboptimality in each round is no more than 2. Notice that the term $J_2$ is equal to the term $I_2$ in the proof of Theorem 4.2 (See (E.1)) and with the same argument, it can be upper bounded by

$$J_2 \leq O\left(\frac{Rd^{1.5}}{\alpha} \times \sqrt{\log^3\left((1+KL^2/\lambda)/\delta\right)} + \frac{dS\sqrt{\lambda}}{\alpha} \times \sqrt{\log^2\left((1+KL^2/\lambda)/\delta\right)}\right.$$

$$\left. + dC\sqrt{\log^2\left((1+KL^2/\lambda)/\delta\right)}\right), \tag{F.2}$$

where the inequality comes from (E.7). For the term $J_1$, we consider for all rounds $k \in [K]$ with $w_k = 1$ and exploration bonus $2\beta\sqrt{\mathbf{x}_k^\top \boldsymbol{\Sigma}_k^{-1}\mathbf{x}_k} \geq \Delta$. For simplicity, we assume these rounds can be listed as $\{k_1, .., k_m\}$. With this notation, for each $i \leq m$, we can construct the auxiliary covariance matrix $\mathbf{A}_i = \lambda\mathbf{I} + \sum_{j=1}^{i-1} \mathbf{x}_{k_j}\mathbf{x}_{k_j}^\top$. Due to the definition of original covariance matrix $\boldsymbol{\Sigma}_k$ in Algorithm (Line 2), we have

$$\boldsymbol{\Sigma}_{k_i} \geq \lambda\mathbf{I} + \sum_{j=1}^{i-1} w_{k_j}\mathbf{x}_{k_j}\mathbf{x}_{k_j}^\top = \mathbf{A}_i.$$

According to Lemma J.4, it further implies that for vector $\mathbf{x}_{k_i}$, we have

$$\mathbf{x}_{k_i}^\top \boldsymbol{\Sigma}_{k_i}^{-1}\mathbf{x}_{k_i} \leq \mathbf{x}_{k_i}^\top (\mathbf{A}_i)^{-1}\mathbf{x}_{k_i}. \tag{F.3}$$

Therefore, the term $J_1$ can be bounded by

$$J_1 = \sum_{k:w_k=1, 2\beta\sqrt{\mathbf{x}_k^\top \boldsymbol{\Sigma}_k^{-1}\mathbf{x}_k} \geq \Delta} \min\left(2, 2\beta\sqrt{\mathbf{x}_k^\top \boldsymbol{\Sigma}_k^{-1}\mathbf{x}_k}\right)$$

$$\leq \sum_{i=1}^{m} 2\beta \min\left(1, \sqrt{\mathbf{x}_{k_i}^\top \boldsymbol{\Sigma}_{k_i}^{-1}\mathbf{x}_{k_i}}\right)$$

$$\leq 2\beta \sum_{i=1}^{m} \min\left(1, \sqrt{\mathbf{x}_{k_i}^\top (\mathbf{A}_i)^{-1}\mathbf{x}_{k_i}}\right)$$

$$\leq 2\beta \sqrt{\sum_{i=1}^{m} 1 \times \sum_{i=1}^{m} \min\left(1, \mathbf{x}_{k_i}^\top (\mathbf{A}_i)^{-1}\mathbf{x}_{k_i}\right)}$$

$$\leq 2\beta\sqrt{2dm\log(1+KL^2/\lambda)}, \tag{F.4}$$

where the first inequality holds since $\beta \geq 1$, the second inequality holds due to (F.3), the third inequality holds due to Cauchy-Schwarz inequality, the fourth inequality holds due to Lemma J.3 with the facts that $m \leq K$ and $\|\mathbf{x}_{k_i}\|_2 \leq L$. On the other hand, the term $J_1$ is lower bounded by

$$J_1 = \sum_{k:w_k=1, 2\beta\sqrt{\mathbf{x}_k^\top \boldsymbol{\Sigma}_k^{-1}\mathbf{x}_k} \geq \Delta} \min\left(2, 2\beta\sqrt{\mathbf{x}_k^\top \boldsymbol{\Sigma}_k^{-1}\mathbf{x}_k}\right) \geq m \times \Delta, \tag{F.5}$$

where the inequality holds due to the definition of $k_i$ with the fact that $\Delta \leq 2$. Combining the upper and lower bound for term $J_1$, we have

$$m \times \Delta \leq 2\beta\sqrt{2dm\log(1 + KL^2/\lambda)},$$

which further implies that

$$m \leq O\big(\beta^2 d\log(1 + KL^2/\lambda)/\mathrm{gap}_{\min}^2\big). \tag{F.6}$$

Substituting the upper bound of $m$ in (F.6) into (F.4), the term $J_1$ can be upper bounded by

$$J_1 \leq O\big(\beta^2 d\log(1 + KL^2/\lambda)/\Delta\big). \tag{F.7}$$

Finally, substituting the upper bounds of term $J_2$ in (F.2) and term $J_1$ in (F.7) into (F.1), the regret can be upper bounded by

$$
\begin{aligned}
\mathrm{Regret}(K) \leq\ & O\big(\beta^2 d\log(1 + KL^2/\lambda)/\Delta\big) + O\bigg(\frac{Rd^{1.5}}{\alpha} \times \sqrt{\log^3\big((1 + KL^2/\lambda)/\delta\big)} \\
& + \frac{dS\sqrt{\lambda}}{\alpha} \times \sqrt{\log^2\big((1 + KL^2/\lambda)/\delta\big)} + dC\sqrt{\log^2\big((1 + KL^2/\lambda)/\delta\big)}\bigg) \\
=\ & O\bigg(R^2 d^2 \log^2\big((1 + KL^2/\lambda)/\delta\big)/\Delta + \frac{\alpha^2 dC^2}{\Delta} \times \sqrt{\log\big(3 + C^2 L^2 K/(R^2\lambda\delta)\big)} \\
& + S^2 d\lambda\log(1 + KL^2/\lambda)/\Delta + \frac{Rd^{1.5}}{\alpha} \times \sqrt{\log^3\big((1 + KL^2/\lambda)/\delta\big)} \\
& + \frac{dS\sqrt{\lambda}}{\alpha} \times \sqrt{\log^2\big((1 + KL^2/\lambda)/\delta\big)} + dC\sqrt{\log^2\big((1 + KL^2/\lambda)/\delta\big)}\bigg).
\end{aligned}
$$

Therefore, we complete the proof of Theorem C.3. $\qquad\square$

## G  Proof of Theorem 4.9

*Proof of Theorem 4.9.* We discuss two cases here.

- For the case $C \leq \bar{C}$, we know that $\bar{C}$ is still a valid upper bound of the corruption level. Thus, CW-OFUL with a $\bar{C}$ corruption level runs successfully, and its regret is upper bounded by $\widetilde{O}(dR\sqrt{K} + d\bar{C}) = \widetilde{O}(dR\sqrt{K} + d\bar{C})$ as Theorem 4.2 suggests.

- For the case $C = \Omega(\bar{C})$, CW-OFUL can not guarantee a sublinear regret. Thus a trivial regret bound (i.e., regret at each round is bounded by 2) applies.

$\qquad\square$

## H  Proof of Theorem 4.12

We introduce our proof of Theorem 4.12, which is adapted from Bogunovic et al. (2021).

*Proof of Theorem 4.12.* In this proof, we consider an arbitrary algorithm satisfying the conditions in the statement of Theorem 4.12, which will run $K$ rounds for any bandit instance. We consider an uncorrupted bandit instance $A_0$ defined as follows. $A_0$ has the decision sets $\mathcal{D}_k = \mathcal{D}$. Here $\mathcal{D} = \{\mathbf{a}_i\}_{1 \leq i \leq d}$, where $\mathbf{a}_i = \mathbf{e}_i$ is the basis in the $d$-dimensional space. Let $\boldsymbol{\theta}_0^* = (1/4, \underbrace{1/8, \ldots, 1/8}_{(d-1)-\text{times}}) \in$

$\mathbb{R}^d$ and $\epsilon_i = 0$. It is easy to see that the optimal policy is to select $\mathbf{a}_1$ at each round, and the regret to select a sub-optimal arm is $1/8$. Since the regret of the algorithm without corruption satisfies $\mathbb{E}\big[\mathrm{Regret}(K)\big] < R_K$, and all the regret comes from selecting $\mathbf{a}_2, \ldots, \mathbf{a}_d$, we have the expected number of rounds to select $\mathbf{a}_2, \ldots, \mathbf{a}_d$ is at most $R_K/(1/8) = 8R_K$. Then by the pigeonhole principle, there exists some $2 \leq i \leq d$ such that the expected number of times to select $\mathbf{a}_i$ is less than $8R_K/(d-1)$. Without loss of generality, we suppose $i = 2$. Then by Markov inequality, with probability at least $1/2$, the number of times to select $\mathbf{a}_2$ is less than $16R_K/(d-1)$.

Next, we consider a corrupted bandit instance $A_1$ defined as follows. $A_1$ has the same decision set $\mathcal{D} = \{\mathbf{e}_i\}$ as $A_0$, while it has a different $\boldsymbol{\theta}_1^* = (1/4, 3/8, \underbrace{1/8, \ldots, 1/8}_{(d-2)-\text{times}})$. $A_1$ is also noiseless, i.e., $\epsilon_i = 0$. Unlike $A_0$, we have an adversary to attack $A_1$ as follows: whenever $\mathbf{a}_2$ is selected and the total corruption level up to the previous step is no more than $4R_K/(d-1) - 1/4$, the adversary corrupts the reward from $3/8$ to $1/8$. Otherwise, the adversary stops to corrupt the reward. With this adversary, the corruption level $C$ is upper bounded by $4R_K/(d-1) - 1/4 + 1/4 = \Omega(R_K/d)$.

For this adversary, since for $A_1$, each selection of $\mathbf{a}_2$ returns a reward $1/8$, then the agent can not tell the difference between $A_0$ and $A_1$ until the total corruption level reaches the threshold $4R_k/(d-1)$ and the adversary stops to corrupt the reward. Therefore, the sequence of rounds for the agent to select $\mathbf{a}_2$ with $A_1$ instance is the same as the sequence for the agent to select $\mathbf{a}_2$ with $A_0$, until the number of rounds to select action $\mathbf{a}_2$ reaches $4R_k/(d-1)/(1/4) = 16R_k/(d-1)$. However, when the total number of times to select $\mathbf{a}_2$ is less than $16R_k/(d-1)$, the agent cannot differentiate $A_0$ and $A_1$ and will follow the same action sequence as $A_0$. In this case, since for $A_1$, $\mathbf{a}_2$ is the optimal action, and all the other actions suffer a $1/8$ regret, then the regret on $A_1$ is at least $1/8 \cdot (K - 16R_K/(d-1)) = \Omega(K)$, where we use the fact that $R_K \leq O(K)$. Therefore, with probability at least $1/2$, the regret is at least $\Omega(K)$, which further implies that the expected regret is lower bounded by $\mathbb{E}[\text{Regret(K)}] \geq 1/2 \times \Omega(K) = \Omega(K)$. Thus, we finish the proof of Theorem 4.12. $\qquad\square$

# I  Proof of Lemmas in Sections 4, D and Appendix E

## I.1  Proof of Lemma 4.1

*Proof of Lemma 4.1.* According to the definition of estimated vector $\boldsymbol{\theta}_k$ in Algorithm 1 (Line 3), we have

$$\boldsymbol{\theta}_k = \boldsymbol{\Sigma}_k^{-1}\mathbf{b}_k = \boldsymbol{\Sigma}_k^{-1}\sum_{i=1}^{k-1} w_i\mathbf{x}_i r_i = \boldsymbol{\Sigma}_k^{-1}\sum_{i=1}^{k-1} w_i\mathbf{x}_i(\mathbf{x}_i^\top\boldsymbol{\theta} + \eta_i + c_i).$$

This equation further implies that the difference between estimated vector $\boldsymbol{\theta}_k$ and the unknown vector $\boldsymbol{\theta}^*$ can be decomposed as:

$$\begin{aligned}
\|\boldsymbol{\theta}_k - \boldsymbol{\theta}^*\|_{\boldsymbol{\Sigma}_k} &= \Big\|\boldsymbol{\Sigma}_k^{-1}\sum_{i=1}^{k-1} w_i\mathbf{x}_i(\mathbf{x}_i^\top\boldsymbol{\theta}^* + \eta_i + c_i) - \boldsymbol{\theta}^*\Big\|_{\boldsymbol{\Sigma}_k} \\
&= \Big\|\boldsymbol{\Sigma}_k^{-1}\sum_{i=1}^{k-1} w_i\mathbf{x}_i(\mathbf{x}_i^\top\boldsymbol{\theta} + \eta_i + c_i) - \boldsymbol{\Sigma}_k^{-1}\Big(\sum_{i=1}^{k-1} w_i\mathbf{x}_i\mathbf{x}_i^\top + \lambda\mathbf{I}\Big)\boldsymbol{\theta}^*\Big\|_{\boldsymbol{\Sigma}_k} \\
&= \Big\|\boldsymbol{\Sigma}_k^{-1}\sum_{i=1}^{k-1} w_i\mathbf{x}_i\eta_i + \boldsymbol{\Sigma}_k^{-1}\sum_{i=1}^{k-1} w_i\mathbf{x}_i c_i - \lambda\boldsymbol{\Sigma}_k^{-1}\boldsymbol{\theta}^*\Big\|_{\boldsymbol{\Sigma}_k} \\
&\leq \underbrace{\Big\|\boldsymbol{\Sigma}_k^{-1}\sum_{i=1}^{k-1} w_i\mathbf{x}_i\eta_i\Big\|_{\boldsymbol{\Sigma}_k}}_{\text{Stochastic error}:I_1} + \underbrace{\Big\|\boldsymbol{\Sigma}_k^{-1}\sum_{i=1}^{k-1} w_i\mathbf{x}_i c_i\Big\|_{\boldsymbol{\Sigma}_k}}_{\text{Corruption error}:I_2} + \underbrace{\Big\|\lambda\boldsymbol{\Sigma}_k^{-1}\boldsymbol{\theta}^*\Big\|_{\boldsymbol{\Sigma}_k}}_{\text{Regularization error}:I_3}, \quad (\text{I.1})
\end{aligned}$$

where the inequality holds due to the fact that $\|\boldsymbol{a} + \boldsymbol{b} + \boldsymbol{c}\|_{\boldsymbol{\Sigma}_k} \leq \|\boldsymbol{a}\|_{\boldsymbol{\Sigma}_k} + \|\boldsymbol{b}\|_{\boldsymbol{\Sigma}_k} + \|\boldsymbol{c}\|_{\boldsymbol{\Sigma}_k}$.

For the stochastic error term $I_1$, it can be bounded by the concentration Lemma J.2 in Abbasi-Yadkori et al. (2011). More specifically, we introduce the auxiliary vector $\mathbf{x}_i'$ and noise $\eta_i'$ such that $\mathbf{x}_i' = \sqrt{w_i}\mathbf{x}_i$ and $\eta_i' = \sqrt{w_i}\eta_i$. According to the definition of weight $\boldsymbol{\theta}_i$ in Algorithm (Line 6), both of these two situations satisfies that the weight $\boldsymbol{\theta}_i$ is bounded by $w_i \leq 1$. Since the original vector $\mathbf{x}_i$ satisfies that $\|\mathbf{x}_i\|_2 \leq L$ and the original stochastic noise $\eta_i$ is $R$-sub Gaussian, these results further imply that

$$\|\mathbf{x}_i'\|_2 = \|\sqrt{w_i}\mathbf{x}_i\|_2 \leq L, \eta_i' = \sqrt{w_i}\eta_i \text{ is } R\text{-sub Gaussian}.$$

With this notation, the covariance matrix $\boldsymbol{\Sigma}_k$ and the stochastic error term $I_1$ can be rewritten and bounded as:

$$\boldsymbol{\Sigma}_k = \lambda\mathbf{I} + \sum_{i=1}^{k-1} w_i\mathbf{x}_i\mathbf{x}_i^\top = \lambda\mathbf{I} + \sum_{i=1}^{k-1} \mathbf{x}_i'(\mathbf{x}_i')^\top$$

$$\begin{aligned}
I_1 &= \left\|\boldsymbol{\Sigma}_k^{-1}\sum_{i=1}^{k-1} w_i\mathbf{x}_i\eta_i\right\|_{\boldsymbol{\Sigma}_k} \\
&= \left\|\sum_{i=1}^{k-1} w_i\mathbf{x}_i\eta_i\right\|_{\boldsymbol{\Sigma}_k^{-1}} \\
&= \left\|\sum_{i=1}^{k-1} \mathbf{x}_i'\eta_i'\right\|_{\boldsymbol{\Sigma}_k^{-1}} \\
&\leq \sqrt{2R^2\log\left(\frac{\det(\boldsymbol{\Sigma}_k)^{1/2}\det(\boldsymbol{\Sigma}_1)^{-1/2}}{\delta}\right)} \\
&\leq R\sqrt{d\log\left((1+KL^2/\lambda)/\delta\right)},
\end{aligned} \tag{I.2}$$

where the first inequality holds due to Lemma J.2 and the second inequality holds due to the facts that $\boldsymbol{\Sigma}_k = \lambda\mathbf{I}_+ \sum_{i=1}^{k-1} \mathbf{x}_i'(\mathbf{x}_i')^\top$ and $\|\mathbf{x}'\|_2 \leq L$.

For the corruption error term $I_2$, it can be bounded by

$$\begin{aligned}
I_2 &= \left\|\boldsymbol{\Sigma}_k^{-1}\sum_{i=1}^{k-1} w_i\mathbf{x}_ic_i\right\|_{\boldsymbol{\Sigma}_k} \\
&= \left\|\boldsymbol{\Sigma}_k^{-1/2}\sum_{i=1}^{k-1} w_i\mathbf{x}_ic_i\right\|_2 \\
&\leq \sum_{i=1}^{k-1} \left\|\boldsymbol{\Sigma}_k^{-1/2}w_i\mathbf{x}_ic_i\right\|_2 \\
&= \sum_{i=1}^{k-1} |c_i| \times w_i\|\boldsymbol{\Sigma}_k^{-1/2}\mathbf{x}_i\| \\
&\leq \sum_{i=1}^{k-1} |c_i|\alpha \\
&\leq \alpha C,
\end{aligned} \tag{I.3}$$

where the first inequality holds due to the fact that $\|\mathbf{a} + \boldsymbol{b}\|_2 \leq \|\mathbf{a}\|_2 + \|\boldsymbol{b}\|_2$, the second inequality holds due to the definition of weight $w_i$ in Algorithm (Line 6) with the fact that $\boldsymbol{\Sigma}_k \succeq \boldsymbol{\Sigma}_i$ and the last inequality holds due to the definition of corruption level $C$.

For the regularization error term $I_3$, we have

$$I_3 = \left\|\lambda\boldsymbol{\Sigma}_k^{-1}\boldsymbol{\theta}^*\right\|_{\boldsymbol{\Sigma}_k} = \lambda\left\|\boldsymbol{\theta}^*\right\|_{\boldsymbol{\Sigma}_k^{-1}} \leq \sqrt{\lambda}\|\boldsymbol{\theta}^*\|_2 \leq \sqrt{\lambda}S, \tag{I.4}$$

where the first inequality holds due to $\left\|\boldsymbol{\theta}^*\right\|_{\boldsymbol{\Sigma}_k} \leq \|\boldsymbol{\theta}^*\|_2/\sqrt{\lambda_{\min}(\boldsymbol{\Sigma}_k)}$ with the fact that $\boldsymbol{\Sigma}_k = \lambda\mathbf{I} + \sum_{i=1}^{k-1} w_i\mathbf{x}_i\mathbf{x}_i^\top \succeq \lambda\mathbf{I}$ and the last inequality holds due to the assumption that $\|\boldsymbol{\theta}^*\|_2 \leq S$.

Finally, substituting the results in (I.2), (I.3) and (I.4) into (I.1), we have

$$\|\boldsymbol{\theta}_k - \boldsymbol{\theta}^*\|_{\boldsymbol{\Sigma}_k} \leq I_1 + I_2 + I_3 \leq R\sqrt{d\log\left((1+KL^2/\lambda)/\delta\right)} + \alpha C + \sqrt{\lambda}S.$$

Therefore, we finish the proof of Lemma 4.1. $\qquad\square$

## I.2 Proof of Lemma E.1

*Proof of Lemma E.1.* Firstly, on the event $\mathcal{E}$, for each round $k \in [K]$ and each action $\mathbf{x} \in \mathcal{D}_k$, we have

$$
\begin{aligned}
\boldsymbol{\theta}_k^\top \mathbf{x} + \beta \sqrt{\mathbf{x}^\top \boldsymbol{\Sigma}_k^{-1} \mathbf{x}} - (\boldsymbol{\theta}^*)^\top \mathbf{x} &= (\boldsymbol{\theta}_k - \boldsymbol{\theta}^*)^\top \mathbf{x} + \beta \sqrt{\mathbf{x}^\top \boldsymbol{\Sigma}_k^{-1} \mathbf{x}} \\
&\geq -\|\boldsymbol{\theta}_k - \boldsymbol{\theta}^*\|_{\boldsymbol{\Sigma}_k} \times \|\mathbf{x}\|_{\boldsymbol{\Sigma}_k^{-1}} + \beta \sqrt{\mathbf{x}^\top \boldsymbol{\Sigma}_k^{-1} \mathbf{x}} \\
&\geq -\beta \|\mathbf{x}\|_{\boldsymbol{\Sigma}_k^{-1}} + \beta \sqrt{\mathbf{x}^\top \boldsymbol{\Sigma}_k^{-1} \mathbf{x}} \\
&= 0, \quad\quad\quad\quad\quad\quad\quad\quad\quad\quad\quad\quad\quad\quad\quad (\text{I.5})
\end{aligned}
$$

where the first inequality holds due to the Cauchy-Schwarz inequality and the last inequality holds due to the definition of $\mathcal{E}$ in Lemma 4.1. (I.5) shows that our estimator in Algorithm 1 is optimistic for each action $\mathbf{x} \in \mathcal{D}_k$. For simplicity, we denote the optimal action at round $k$ as $\mathbf{x}^* = \arg\max_{\mathbf{x} \in \mathcal{D}_k} (\boldsymbol{\theta}^*)^\top \mathbf{x}$ and (I.5) further implies that the regret at round $k$ can be upper bounded by

$$
\begin{aligned}
\Delta_k &= (\boldsymbol{\theta}^*)^\top \mathbf{x}^* - (\boldsymbol{\theta}^*)^\top \mathbf{x}_k \\
&\leq \boldsymbol{\theta}_k^\top \mathbf{x}^* + \beta \sqrt{(\mathbf{x}^*)^\top \boldsymbol{\Sigma}_k^{-1} \mathbf{x}^*} - (\boldsymbol{\theta}^*)^\top \mathbf{x}_k \\
&\leq \boldsymbol{\theta}_k^\top \mathbf{x}_k + \beta \sqrt{\mathbf{x}_k^\top \boldsymbol{\Sigma}_k^{-1} \mathbf{x}_k} - (\boldsymbol{\theta}^*)^\top \mathbf{x}_k \\
&= (\boldsymbol{\theta}_k - \boldsymbol{\theta}^*)^\top \mathbf{x}_k + \beta \sqrt{\mathbf{x}_k^\top \boldsymbol{\Sigma}_k^{-1} \mathbf{x}_k} \\
&\leq \|\boldsymbol{\theta}_k - \boldsymbol{\theta}^*\|_{\boldsymbol{\Sigma}_k} \times \|\mathbf{x}_k\|_{\boldsymbol{\Sigma}_k^{-1}} + \beta \sqrt{\mathbf{x}_k^\top \boldsymbol{\Sigma}_k^{-1} \mathbf{x}_k} \\
&\leq 2\beta \sqrt{\mathbf{x}_k^\top \boldsymbol{\Sigma}_k^{-1} \mathbf{x}_k},
\end{aligned}
$$

where the first inequality holds due to (I.5), the second inequality holds due to the selection rule in Algorithm (Line 5), the third inequality holds due to the Cauchy-Schwarz inequality and the last inequality holds due to the definition of $\mathcal{E}$ in Lemma 4.1. Thus, we finish the proof of Lemma E.1. $\qquad\square$

## J Auxiliary Lemmas

**Lemma J.1** (Azuma–Hoeffding inequality, Cesa-Bianchi and Lugosi 2006). Let $\{\eta_k\}_{k=1}^K$ be a martingale difference sequence with respect to a filtration $\{\mathcal{G}_k\}$ satisfying $|\eta_k| \leq R$ for some constant $R$, $\eta_k$ is $\mathcal{G}_{k+1}$-measurable, $\mathbb{E}[\eta_k | \mathcal{G}_k] = 0$. Then for any $0 < \delta < 1$, with high probability at least $1 - \delta$, we have

$$
\sum_{k=1}^K \eta_k \leq R \sqrt{2K \log(1/\delta)}.
$$

**Lemma J.2** (Lemma 9 in Abbasi-Yadkori et al. 2011). Let $\{\epsilon_k\}_{k=1}^K$ be a real-valued stochastic process with corresponding filtration $\{\mathcal{F}_k\}_{k=0}^K$ such that $\epsilon_k$ is $\mathcal{F}_k$-measure and $\epsilon_k$ is conditionally $R$-sub-Gaussian, *i.e.*

$$
\forall \lambda \in \mathbb{R}, \mathbb{E}[e^{\lambda \epsilon_k} | \mathcal{F}_{k-1}] \leq \exp\left(\frac{\lambda^2 R^2}{2}\right).
$$

Let $\{\mathbf{x}_k\}_{k=1}^K$ be an $\mathbb{R}^d$-valued stochastic process where $\mathbf{x}_k$ is $\mathcal{F}_{k-1}$-measurable and for any $k \in [K]$, we further define $\boldsymbol{\Sigma}_k = \lambda \mathbf{I} + \sum_{i=1}^k \mathbf{x}_i \mathbf{x}_i^\top$. Then with probability at least $1 - \delta$, for all $k \in [K]$, we have

$$
\left\| \sum_{i=1}^k \mathbf{x}_i \eta_i \right\|_{\boldsymbol{\Sigma}_k^{-1}}^2 \leq 2R^2 \log\left( \frac{\det(\boldsymbol{\Sigma}_k)^{1/2} \det(\boldsymbol{\Sigma}_0)^{-1/2}}{\delta} \right).
$$

**Lemma J.3** (Lemma 11 in Abbasi-Yadkori et al. 2011)**.** Let $\{\mathbf{x}_k\}_{k=1}^K$ be a sequence of vectors in $\mathbb{R}^d$, matrix $\boldsymbol{\Sigma}_0$ a $d \times d$ positive definite matrix and define $\boldsymbol{\Sigma}_k = \boldsymbol{\Sigma}_0 + \sum_{i=1}^k \mathbf{x}_i \mathbf{x}_i^\top$, then we have

$$\sum_{i=1}^k \min \left\{ 1, \mathbf{x}_i^\top \boldsymbol{\Sigma}_{i-1}^{-1} \mathbf{x}_i \right\} \le 2 \log \left( \frac{\det \boldsymbol{\Sigma}_k}{\det \boldsymbol{\Sigma}_0} \right).$$

In addition, if $\|\mathbf{x}_i\|_2 \le L$ holds for all $i \in [K]$, then

$$\sum_{i=1}^k \min \left\{ 1, \mathbf{x}_i^\top \boldsymbol{\Sigma}_{i-1}^{-1} \mathbf{x}_i \right\} \le 2 \log \left( \frac{\det \boldsymbol{\Sigma}_k}{\det \boldsymbol{\Sigma}_0} \right) \le 2 \left( d \log \left( (\text{trace}(\boldsymbol{\Sigma}_0) + kL^2)/d \right) - \log \det \boldsymbol{\Sigma}_0 \right).$$

**Lemma J.4** (Corollary 7.7.4. (a) in Horn and Johnson 2012)**.** Let $\mathbf{A}, \mathbf{B}$ be a Hermitian matrix in $\mathbb{R}^{d \times d}$ and suppose $\mathbf{A}, \mathbf{B} \succ \mathbf{0}$, then $\mathbf{A} \succeq \mathbf{B}$ if and only if $\mathbf{B}^{-1} \succeq \mathbf{A}^{-1}$.