# OpenReview forum: "Nearly Optimal Algorithms for Linear Contextual Bandits with Adversarial Corruptions"
_NeurIPS.cc/2022/Conference — NeurIPS 2022 Accept_

### Official Review · Reviewer_ka6i · 2022-07-11

**Rating:** 6
**Confidence:** 4
**Soundness:** 3 good
**Presentation:** 2 fair
**Contribution:** 3 good

**Summary:**

This paper studies the problem of linear contextual bandits with adversarial corruptions. They focus on the setting with a powerful adversary who attacks after seeing the learner’s action. The authors propose a weighted-OFUL algorithm for this setting, which is also the first to use weighted least square for solving adversarial corruption problems. They derive the near-optimal \tilde{O}(d\sqrt{K}+dC) regret upper bound for the C-known setting. And for the unknown setting, they show O(d\sqrt{K}) regret when C\le \sqrt{T} and \Theta(T) regret when C>\sqrt{T}.

**Questions:**

Can you incorporate more discussions on how the current analysis dealt with the powerful adversary and why previous works fail? For example, the algorithm in Zhao et al. is also weighted OFUL with enlarged confidence radius. What is the key difference that make their algorithm work with a weaker adversary?

**Limitations:**

The work is about online learning theory and does not have negative societal impact.

**Strengths And Weaknesses:**

Strength:

This paper studies the linear contextual bandit problem with potential infinite arms and dynamic action sets. Their proposed algorithm is a novel attempt using the previously well-known weighted least square technique to solve the corruption problem.
The derived results show an advantage over previous works. When C is known, they provide a near-optimal upper bound of order O(d\sqrt{T}+dC), which matches the lower bound in both the uncorrupted setting and corruption setting. This result also improves previous results in the corruption setting. When C is unknown, they show that the regret is Theta(d\sqrt{T}) when C\le \sqrt{T} and is \Theta(T) otherwise.

Weakness:

There are too many remarks in the paper. And the presentation in each remark is not easy to follow. I think a clearer way to compare with previous works would be to complete the table to incorporate more comparable information, e.g, powerful adversary, computation efficiency, (in)finite (fixed) action set, (un)known C and its value range, the value of Cr and C’, etc.
In stochastic linear bandits, the O(d\sqrtT+C) regret derived by Chung-Wei Lee et al [Achieving Near Instance-Optimality and Minimax-Optimality in Stochastic and Adversarial Linear Bandits Simultaneously, ICML2021] is also comparable.

Detailed comments:

Line 123: D_k\in R^d should be D_k\subseteq R^d

Line 134 equation: what is \epsilon_{1:k-1}?

Line 198: estimated

Line 40-46: Li et al., 2010 can also handle infinite arm case.

---

> ### Author Response · Authors · 2022-08-02
> **Response to Reviewer ka6i**
>
> Thank you for your encouraging comments. We address them as follows.
>
> **Q1**: There are too many remarks in the paper. And the presentation in each remark is not easy to follow.
>
> **A1**: Thanks for your suggestion. We have reorganized Section 4 in the revision and also simplified some remarks for a better presentation flow.
> ***
> **Q2**: I think a clearer way to compare with previous works would be to complete the table to incorporate more comparable information, e.g, powerful adversary, computation efficiency, (in)finite (fixed) action set, (un)known C and its value range, the value of Cr and C’, etc.
>
> **A2**: Thank you for your great suggestion. We have added the information of the powerful adversary (strong vs weak adversary), computation efficiency, and (un)known corruption level C in Table 1 in the revised paper. Since all algorithms in the table focus on the infinite unfixed action set and have the same value range, we do not add the (in)finite (fixed) action set and value range information into the table.
> ***
>
> **Q3**: In stochastic linear bandits, the O(d\sqrtT+C) regret derived by Chung-Wei Lee et al [Achieving Near Instance-Optimality and Minimax-Optimality in Stochastic and Adversarial Linear Bandits Simultaneously, ICML2021] is also comparable.
>
> **A3**: Our work and previous results in Table 1 focus on the contextual linear bandit problem where the action set can vary across different rounds. In contrast, Lee et al. (2021) focused on stochastic linear bandits, where the action set is fixed. Our setting is more general than that of Lee et al. (2021).
> In addition, Lee et al. (2021) also needs an additional assumption that adversarial corruptions are generated through the inner product of an adversarial vector and the contextual vector, while we make no additional assumption in our paper. In the revised paper, we have added a discussion on these differences to the related work part “Bandits with Adversarial Rewards”.
>
> ***
>
> **Q4**: Some typos
>
> **A4**: We’re sorry for the typos. We have fixed them in the revision.
> ***
> **Q5**: Can you incorporate more discussions on how the current analysis dealt with the powerful adversary and why previous works fail? For example, the algorithm in Zhao et al. is also weighted OFUL with enlarged confidence radius. What is the key difference that make their algorithm work with a weaker adversary?
>
> **A5**: The key difference between the weak adversary and the powerful adversary is that, the weak adversary must corrupt the rewards before the agent selects its actions, while the powerful adversary (i.e., strong adversary) can corrupt the rewards after seeing the action being selected by the agent. Therefore, for the weak adversary, one can use the randomness to ‘counter’ the adversary. For instance, the Multi-level weighted OFUL (Zhao et al., 2022) randomly selects one of the individual learners with certain probability at each round, and it may select some learner who is less affected by the adversary. In contrast, a powerful adversary (strong adversary) can see the agent’s actions, which makes the randomization-based approach fail to defend against the adversary.

---

> > ### Comment · Reviewer_ka6i · 2022-08-05
> > **Still have concerns about the stronger adversary**
> >
> > Thanks for your reply. I still have concerns about the technique to deal with a stronger adversary. Suppose we only consider the case of known corruption level C. In this case, Zhao et al., [Linear Contextual Bandits with Adversarial Corruptions] use a deterministic weighted OFU algorithm with an enlarged confidence interval, which is very similar to your algorithm. My question is what is the key analysis difference that makes your algorithm work with a stronger adversary? Or does their analysis also defend a stronger adversary though written w.r.t a weak one?

---

> > > ### Author Response · Authors · 2022-08-06
> > > **Reply to 'Still have concerns about the stronger adversary'**
> > >
> > > Thank you for your follow-up questions.
> > >
> > > **Q1**: Suppose we only consider the case of known corruption level C. In this case, Zhao et al., [Linear Contextual Bandits with Adversarial Corruptions] use a deterministic weighted OFU algorithm with an enlarged confidence interval, which is very similar to your algorithm.
> > >
> > > **A1**: We would like to highlight the differences between the RobustOFUL in Zhao et al. (2022) and our CW-OFUL even when the corruption level C is known.
> > > It is true that both RobustOFUL and CW-OFUL adapt a weighted regression scheme to estimate the underlying vector $\theta^*$. However, it is worth noting that Zhao et al. (2022) consider a heteroscedastic corrupted bandit setting where the variances of rewards at different time steps are different, and the weight used in RobustOFUL is $\max(\alpha, \sigma_t)$, where $\alpha$ is a constant and $\sigma_t$ is the variance of the reward at step $t$. With the help of such variance-dependent weights, Zhao et al. (2022) can obtain a variance-aware regret guarantee in their Theorem 4.2. In contrast, we consider the vanilla corrupted bandit setting (the variances of rewards at different steps are identical), and the weight used in our CW-OFUL is $\max(\alpha, \\|x\\|_{\Sigma^{-1}})$,
> > >
> > > which depends on the ‘uncertainty’ quantity $\\|x\\|_{\Sigma^{-1}}$. The uncertainty-dependent weights can reduce the confidence radius from $O(C\sqrt{d})$ (Lemma 4.1 in Zhao et al. (2022) ) to $O(\sqrt{d})$ and obtain a near-optimal regret when the corruption level is known. This is also explained in Section 3.2 of our paper. Therefore, although both Zhao et al. (2022) and us consider weighted linear regression, the design of weights, analyses and the corresponding theoretical results (e.g., confidence set radius, regret) are rather different.
> > >
> > >
> > > ***
> > >
> > > **Q2**: My question is what is the key analysis difference that makes your algorithm work with a stronger adversary? Or does their analysis also defend a stronger adversary though written w.r.t a weak one?
> > >
> > > **A2**: For the case where the corruption level C is known to the agent, indeed both the RobustOFUL in Zhao et al. (2022) and our CW-OFUL can deal with the stronger adversary. As we explained in **A1**, the main difference is the design of the weight in linear regression, which affects the subsequent analysis and confidence radius, and eventually the regret.
> > >
> > > When the corruption level is unknown, the Multi-level weighted OFUL algorithm in Zhao et al. (2022) cannot deal with a strong adversary. In particular, in the presence of a strong adversary, Lemma 7.2 and Theorem 6.1 about Multi-level weighted OFUL in Zhao et al. (2022) no longer hold due to the following reason. The key step in Multi-level weighted OFUL is to randomly select one of the individual learners with a certain probability at each round (See Line 11 of their algorithm), and they show that some learner will be less affected by the adversary, e.g., Lemma 7.2 shows that the corruption level for the learner at level $l^*=\max(2,\log_2 C)$ is at most $\tilde{O}(1)$. Note that the ‘less affected learner’ only exists when the adversary is a weaker adversary, since the corruption of each arm by the weak adversary occurs before the agent makes its decision, thus the adversary cannot make the agent suffer from a large corruption thanks to the random level selection scheme. However, for a stronger adversary, it can always make the corruption for the learner at level $l^*$ up to $O(C)$ after the agent selects the level, which makes their Lemma 7.2 and their final regret guarantee (Theorem 6.1) no longer hold.
> > > In contrast, our CW-OFUL is a deterministic algorithm, and it does not matter if the adversary makes the corruption before or after the agent selects the arm. This makes our algorithm and analysis hold for both the weaker adversary and the strong adversary.
> > >
> > > Please let us know if you have any further questions. Thanks.

---

> > > > ### Comment · Reviewer_ka6i · 2022-08-07
> > > > **Thanks for your detailed explanations**
> > > >
> > > > I have no concerns anymore.

---

> > > > > ### Author Response · Authors · 2022-08-07
> > > > > **Thank you for your positive feedback!**
> > > > >
> > > > > We're glad that we have fully addressed your concerns. Thank you again for all the suggestions for improving the presentation of our paper. We would appreciate it if you could consider raising your score. Thank you!

---

### Official Review · Reviewer_MF5t · 2022-07-11

**Rating:** 6
**Confidence:** 3
**Soundness:** 3 good
**Presentation:** 3 good
**Contribution:** 3 good

**Summary:**

This study considers linear contextual bandits with adversarial corruption and proposes a UCB-based algorithm.
When the corruption level $C$ is the proposed algorithm with a tuned hyperparameter achieves a tight regret bound depending on $C$.
For the case of unknown $C$, the algorithm achieves a nearly optimal bound if $C$ is at most the square root of the number of rounds.
This result turns out to be in some sense the best theoretical guarantee, which follows from a newly proven lower bound.


**Questions:**


- What exactly does "computationally efficient" mean? Even when the decision set is a convex set as in Proposition 4.7., I don't think it is clear whether the optimization problem in Step 5 of Algorithm 1 can be solved in polynomial time as the objective function is nonconvex. The case of exponentially large decision set (e.g., $D_k = \{-1  , 1 \}^d$) is also non-trivial.

- Can we obtain improved regret bounds for the case in which the size of $D_k$ is bounded? (e.g., regret bound depending on $\sqrt{d \log |D_k| }$ instead of $d$?)

**Limitations:**

The limitations are adequately addressed.

**Strengths And Weaknesses:**

Strengths:

- Regret bounds are nearly tight.
- The proposed algorithm is simple and computationally efficient.

Weaknesses:

- The "optimality" of the algorithm in the case where $C$ is unknown is somewhat limited. In fact, the algorithm is nearly optimal among \textit{algorithms that achieve a near-optimal regret bound for uncorrupted cases}.
- There is little novelty in the algorithm and analysis.

Comments:

I believe that the results are sufficiently impactful, as they provide a nearly tight regret bound for the important problem of contextual linear bandit with adversarial corruption.
The proposed algorithm is surprisingly simple compared to that of existing studies, and appears to be practical.

The description of lower bound for unknown C in Table 1 ($\Omega(K), C \geq \sqrt{K}$) may be misleading.
What we can actually say would be that: any algorithm that achieves $O(d \sqrt{K})$-regret for $C = 0$ suffers $\Omega(K)$-regret for some $C \geq \sqrt{K}$ in the worst case.
I don't think this conditioning of the algorithm should be omitted.


Line 19: an decision set <- a decision set

Line 266: a environment parameter <- an environment parameter

Supplementary (G.4): $\lambda \| theta^* \|_{\Sigma_k}$
 <-
$\lambda \| theta^* \|_{\Sigma_k}^{-1}$

---

> ### Author Response · Authors · 2022-08-02
> **Response to Reviewer MF5T**
>
> Thank you for your positive comments. We address them as follows.
>
> **Q1**: There is little novelty in the algorithm and analysis
>
> **A1**: We respectfully disagree with the reviewer’s evaluation on the novelty of our algorithm and analysis. Firstly, our algorithm is simple but also novel. At the core of the CW-OFUL algorithm is a weighted ridge regression where the weight of each chosen arm is adaptive to its ‘uncertainty’ ($|x|_{\Sigma^{-1}}$). Compared with previous algorithms such as OFUL, we are the first to use uncertainty as a weight in linear regression and to show that such uncertainty-dependent weights can deal with adversary corruption very effectively. Though the analysis largely follows standard techniques, we do not think it is a shortcoming since it further emphasizes the novelty of our algorithm design. Note that before our work, there were many attempts to achieve optimality in corruption-robust linear contextual bandits, but none of them provides an optimal solution even with very complicated algorithm design. So we believe our algorithm is very novel and significant (though it is very simple, which is also an advantage).
>
> ***
>
> **Q2**:  The description of lower bound for unknown C in Table 1 may be misleading.
>
> **A2**: We guess the reviewer may overlook footnote 4 of the table. In the footnote, we have already explained the requirement of the lower bound and mentioned that it is possible for an algorithm that does not achieve the optimal regret for uncorrupted bandits (e.g., $R_K = O(K^{0.75})$) to achieve a sub-linear regret in the presence of corruptions.
>
> ***
>
> **Q3**: Some typos
>
> **A3**: Thank you for pointing them out. We have fixed them in the revision.
>
> ***
>
> **Q4**: Questions about the "computationally efficient"
>
> **A4**: In this work, we assume there is a computation oracle to solve the linear optimization problems over the decision set $\mathcal{D_t}$ (e.g., Line 5 of Algorithm 1). This is implicitly assumed in almost all existing works for solving contextual linear bandit problems with infinite arms (e.g., OFUL and LinUCB algorithms); otherwise, choosing an arm from the infinite decision set is computationally intractable. In the special case that the decision set is finite, such a computation oracle apparently exists. Meanwhile, it is worth noting that Robust VOFUL and COBE+VOFUL Algorithms are still computationally intractable even with the computation oracle over the decision set, since they need to solve a maximization problem over a nonconvex confidence set, which is defined as the intersection of an exponential number of sets. We have added a footnote in Table 1 to explicitly define computational efficiency.

---

### Official Review · Reviewer_vcvT · 2022-07-12

**Rating:** 6
**Confidence:** 3
**Soundness:** 4 excellent
**Presentation:** 3 good
**Contribution:** 4 excellent

**Summary:**

The authors consider the problem of linear contextual bandits with adversarial corruptions. In this framework, at each iteration k of the game, the learner has to select one action x from a decision set D_k.  The environment then generates a reward based on that action, an unknown environment parameter and some stochastic noise. The adversary observes the rewards and then decides how much corruption to add to it using the knowledge of the action set, the chosen action, and the stochastic reward.  This definition of corruption is similar to the one in the recent work of Bogunovic et al. (2021), but prior works (in particular Lykouris et al. (2019)) consider a model where the adversary sets the corruption before the learner chooses an action.
In this work, the authors consider both the settings with known and unknown corruption levels and derive algorithms which are optimal up to logarithmic factors, improving the state of the art in both cases. In the case with known corruption level, prior works could be suboptimal due to different factors: the corruption level was either a mutliplicative factor in the regret bound, giving a term that depends on the product of the total corruption and the square root of the number of iterations, or, in the work of Wei et al. (2022) ifthe corruption level was an additive terms, then the algorithm suffered a worst dependency in the dimension of the context $d$ as well as being computationally inefficient.
In contrast, the current approach achieves the correct dependency on $d$ and on $C$ up to logarithmic factors without being as computationally expensive.

 In the case with unknown corruption level, they notably complete the analysis of the problem by providing a lower bound for problems that have a large corruption level ($C >= \sqrt K$) and derive upper bounds for both small and large values of corruptions, matching up to logarithmic factors when the level of corruption is small and up to constants when it is large, as the bound is in $\theta (K)$, making it impossible to achieve sublinear regret.
Importantly, this result is robust to a larger amount of corruptions than some of the previous results (Zhao et al. 2021), and similarly to the case with known corruption level, it achieves the correct dependency on $d$ and on $C$.

The method is based on the OFUL framework (optimism in face of uncertainty), which is similar to the rest of the litterature, and uses  weighted ridge regression with a novel choice of weights, which are based on the inverse of an exploration parameter that is trunctated to provide stability and robustness against noise and adversarial corruption.

**Questions:**

Section 4 could be rephrased to improve clarity and readability.
In particular:
- in section 4.1, paragrpah titles could benefit from being more highlighted than the remarks title  (see Lower bound line 262 for example).
- the phrasing of line 292 in theorem 4.9 is not very intuitive.
- remark 4.13 seems to be a proof sketch of theorem 4.12, and could be identified as such.

**Strengths And Weaknesses:**

This paper follows a long line of works on the problem of linear contextual bandits with adversarial corruptions, and provides substantial improvements by being the first capable to match lower bounds up to logarithmic factor, improving upon the dependency on the corruption level or the dimension of the context.  The results are significant, yet the proposed algorithm is elegant and relatively simple, and the proofs look sound.
It is nice that problem dependent type of bounds are also presented in the appendix.

The paper is well written, with a particular focus on relating their results with the rest of the litterature.
I would still point out that section 4 is really dense, making it somewhat difficult to identify the key results. The remarks in the section are important and should not be removed, but there may be a way to restructure that section such that the results of this work are easier to identify.

---

> ### Author Response · Authors · 2022-08-02
> **Response to Reviewer vcvT**
>
> Thank you for your positive comment!
>
> **Q1**: Section 4 could be rephrased to improve clarity and readability.
>
> **A1**: Thank you for your suggestion. We have reorganized Section 4 in the revision for better readability.

---

> > ### Comment · Reviewer_vcvT · 2022-08-07
> > **Thank you for your answer**
> >
> > Hi,
> >
> > Thank you for your answer, I do not have further questions, the other reviewers already mentioned many interesting things.

---

### Official Review · Reviewer_Ccme · 2022-07-12

**Rating:** 6
**Confidence:** 4
**Soundness:** 3 good
**Presentation:** 3 good
**Contribution:** 2 fair

**Summary:**

The paper considers the linear bandits setup with adversarial corruptions. Although the problem is well studied, the authors propose a variant of the well known OFUL algorithm, where the weights are chosen in a specific manner. It is argued that if the corruption level is known, giving weight inverse to the confidence radius makes the algorithm robust. The analysis is done in both settings where the level of corruption is known and unknown. Appropriate lower bounds are provided to support the claims.

**Questions:**

See the weakness part

Thanks to the authors for providing a convincing response. I increase my score to 6 now.

**Strengths And Weaknesses:**

Strengths:
Writing: The paper is well written and easy to follow. The short calculations are quite helpful.

Related work and comparison: The known C part is well written and compared with the previous literature quite exhaustively.

Weakness:
I think the main weakness of the paper is the setup where the corruption level is not known. What is the algorithm here? How will I estimate \bar{C}? If I overestimate, I would incur linear regret, and so it is of importance to know how this estimation is done.

In practical situations, the total level of corruption won’t be provided in advance, and so an algorithm must estimate it in a structured manner, not leave it as tuning parameter. I know some previous work have done so, but that seems incomplete--although a lot of non trivial unknown C papers are written, as the author themselves refer. Please comment on this.

Validation: It would be great if the weighted OFUL can be implemented, even in a simple finite arm setting (or using the tricks, via Thompson Sampling etc, Russo and Van Roy)

---

> ### Author Response · Authors · 2022-08-02
> **Response to Reviewer Ccme**
>
> Thank you for your comments. We address them as follows.
>
> **Q1**: I think the main weakness of the paper is the setup where the corruption level is not known. What is the algorithm here? How will I estimate \bar{C}? If I overestimate, I would incur linear regret, and so it is of importance to know how this estimation is done.
>
> **A1**: We apologize for not making it clear. Here we introduce \bar{C} just for the purpose of generality. In fact, when the corruption level C is unknown, we can choose \bar{C} = \sqrt{K}, as we indicated in Line 295 (the last part of Theorem 4.9). There is no need to estimate the corruption level C. That is why we did not provide a new algorithm description here. It is essential the same algorithm as Algorithm 1, except that we choose \bar{C} = \sqrt{K}.  As you can see, Theorem 4.9 shows that the upper bound of our algorithm (Algorithm 1) when choosing \bar{C} = \sqrt{K} is O(d\sqrt{K}). This together with the lower bound result in Theorem 4.12, suggests that this is the best result one can achieve when the corruption level C is unknown.
>
> ***
> **Q2**: In practical situations, the total level of corruption won’t be provided in advance, and so an algorithm must estimate it in a structured manner, not leave it as tuning parameter.
>
> **A2**. As we have explained above, our algorithm does not need to estimate C when C is unknown. Setting \bar{C} = \sqrt{K} in Algorithm 1 achieves the optimal regret.
> ***
>
> **Q3**: It would be great if the weighted OFUL can be implemented, even in a simple finite arm setting
>
> **A3**. Thanks for your suggestion! We have conducted simulations on synthetic datasets to compare our algorithm with the Greedy algorithm in Bogunovic et al. (2021) with unknown corruption level C. The settings and results are reported in Appendix B in the revised paper.
>
> In detail, we construct a linear bandit instance with the dimension $d=5$ and the true model parameter $\theta^*=[1/\sqrt{d},...,1/\sqrt{d}]$. During each round $k \in [K]$, the decision set $\mathcal{D}_k$ consists of 20 different actions and each action is uniformly sampled from the space $[+1/\sqrt{d},-1/\sqrt{d}]^d$, which satisfies the positive minimum eigenvalue assumption for greedy algorithm. For the attack method, we choose the flip-$\theta$ attack. More specifically, with a corruption level of $C$, the adversary tricks the learner by flipping the value, i.e., $r_t(x_k)=- \langle x_k,\theta^* \rangle$ in the first $C$ rounds. Then, the agent will receive the final reward with Gaussian noise (R=0.1).
>
> In our experiments, we run a simulation with the number of rounds K=10000 and corruption levels from 0,50,100,...,400 to 450 (10 different corruption levels). We apply our CW-OFUL algorithm and compare its performance with Greedy (Bogunovic et al. 2021) and OFUL (Abbasi-Yadk 2011).  We repeat each algorithm 100 times and report their averaged results.
>
> Here are the results of the final regret with different corruption levels.
>
> | Algorithm\ Corruption level |   0   |  50 |  100|     150  | 200      | 250   | 300      | 350   |400      | 450   |
> |:-------------:    |:---------: |:--------:|:--------:|:---------:   |:------:       |:--------:  |:------:       |:--------:  |:------:       |:--------:  |
> |OFUL   |     9.5    |  140.0    | 278.2    | 423.0    | 562.0 | 707.1| 849.8| 990.8 | 1136.8 | 1276.5|
> |Greedy                |   3.7 | 155.6   |  307.3     |465.4     |625.5  |782.6 | 938.1| 1099.8 | 1264.0 | 1413.7 |
> |CW-OFUL        |     21.8       |  103.5    |  199.8  | 313.2   |  438.9  | 571.2| 711.0 | 834.9 | 980.0 | 1110.8|
>
> These results suggest that our CW-OFUL algorithm outperforms both the Greedy and OFUL when the corruption level is larger than 50. In addition, the regret of all tested algorithms demonstrates an additive linear dependence on the corruption level $C$, which corroborates our theoretical results.

---

> ### Author Response · Authors · 2022-08-07
> **Follow-up**
>
> Dear Reviewer,
>
> Since the deadline for the author-reviewer discussion phase is fast approaching, we would like to follow up to hear your feedback on our response. In our rebuttal, we have addressed all your questions. According to your suggestion, we have added numerical experiments to validate our CW-OFUL algorithm and compare it with several baselines. Please let us know if you have any further questions or suggestions. Thank you!
>
> Best,
> Authors

---

### Meta-Review · Area_Chair_mEPS · 2022-08-26

**Recommendation:** Accept
**Confidence:** Certain

**Metareview:**

The paper was generally well-received by the reviewers, who appreciated the contributions, especially the tightness of the bounds and the simplicity of the algorithm. The minor concerns raised in the original reviews were all addressed in the rebuttal, and eventually all reviewers agreed that the paper is suitable for publication at NeurIPS 2022. The authors are encouraged to take all the reviewers' comments into account when preparing the final version of the paper.

One remaining technical concern that needs to be clarified in the final version is the computational efficiency of the proposed algorithm, as raised by reviewers MF5t and Ccme. In their response to reviewer MF5t, the authors erroneously claimed that a linear optimization oracle suffices to implement their algorithm, even though the objective function maximized in the implementation is a nonlinear, strictly convex function. However, the challenge of solving such potentially intractable optimization problems is not unique to this particular algorithm, and indeed all OFUL / LinUCB style methods require solving similar problems --- see, e.g., the discussion in Section 19.3.1. in "Bandit Algorithms" by Lattimore and Szepesvari. The final version should correct the confusing claims about this computational issue, for example by simply updating footnote 4 in the current draft.

**Award:**

No

---

### Decision · Program_Chairs · 2022-09-14

Accept